# HRS phosphorylation drives immunosuppressive exosome secretion and restricts CD8+ T-cell infiltration into tumors

Lei Guan[1,12], Bin Wu[1,12], Ting Li[2,12], Lynn A. Beer[3], Gaurav Sharma[4], Mingyue Li[2], Chin Nien Lee[2], Shujing Liu[2], Changsong Yang[1], Lili Huang[2], Dennie T. Frederick [5], Genevieve M. Boland[6], Guangcan Shao[7], Tatyana M. Svitkina [1], Kathy Q. Cai[8], Fangping Chen[9], Meng-Qiu Dong [7], Gordon B. Mills [10], Lynn M. Schuchter[4,11], Giorgos C. Karakousis[2], Tara C. Mitchell [4,11], Keith T. Flaherty [5], David W. Speicher[3], Youhai H. Chen [2], Meenhard Herlyn [3], Ravi K. Amaravadi[4], Xiaowei Xu[2] & Wei Guo [1✉]

The lack of tumor infiltration by CD8+ T cells is associated with poor patient response to anti-PD-1 therapy. Understanding how tumor infiltration is regulated is key to improving treatment efficacy. Here, we report that phosphorylation of HRS, a pivotal component of the ESCRT complex involved in exosome biogenesis, restricts tumor infiltration of cytolytic CD8+ T cells. Following ERK-mediated phosphorylation, HRS interacts with and mediates the selective loading of PD-L1 to exosomes, which inhibits the migration of CD8+ T cells into tumors. In tissue samples from patients with melanoma, CD8+ T cells are excluded from the regions where tumor cells contain high levels of phosphorylated HRS. In murine tumor models, overexpression of phosphorylated HRS increases resistance to anti-PD-1 treatment, whereas inhibition of HRS phosphorylation enhances treatment efficacy. Our study reveals a mechanism by which phosphorylation of HRS in tumor cells regulates anti-tumor immunity by inducing PD-L1+ immunosuppressive exosomes, and suggests HRS phosphorylation blockade as a potential strategy to improve the efficacy of cancer immunotherapy.

[1] Department of Biology, School of Arts & Sciences, University of Pennsylvania, Philadelphia, PA 19104, USA. [2] Department of Pathology and Laboratory Medicine, Perelman School of Medicine, University of Pennsylvania, Philadelphia, PA 19104, USA. [3] Molecular & Cellular Oncogenesis Program, Wistar Institute, Philadelphia, PA 19104, USA. [4] Abramson Cancer Center, Perelman School of Medicine, University of Pennsylvania, Philadelphia, PA 19104, USA. [5] Division of Medical Oncology, Department of Medicine, Massachusetts General Hospital Cancer Center, Harvard Medical School, Boston, MA 02114, USA. [6] Department of Surgical Oncology, Massachusetts General Hospital, Boston, MA MA02114, USA. [7] National Institute of Biological Sciences, Beijing 102206, P. R. China. [8] Histopathology Facility, Fox Chase Cancer Center, Philadelphia, PA 19111, USA. [9] Histotechnology Facility, The Wistar Institute, Philadelphia, PA 19104, USA. [10] Division of Oncological Science, School of Medicine and Knight Cancer Institute, Oregon Health & Science University, Portland, OR 97201, USA. [11] Department of Medicine, Perelman School of Medicine, University of Pennsylvania, Philadelphia, PA 19104, USA. [12] These authors contributed equally: Lei Guan, Bin Wu, Ting Li. ✉email: guowei@sas.upenn.edu

mmune checkpoint blockade (ICB) using anti-programmed death protein 1 (PD-1) antibodies has demonstrated efficacy in the treatment of many types of cancers[1–3]. Despite this remarkable progress, the majority of patients do not respond to ICB therapies. Recent studies indicate that patients with high intratumoral, but not peritumoral, CD8[+] T cells, have a better response to ICB[4–8]. To improve the efficacy of immunotherapy, it is imperative to promote intratumoral migration of cytolytic CD8[+] T cells[9,10]. However, the molecular mechanisms that regulate tumor infiltration by CD8[+] T cells remain unclear.

Exosomes are small extracellular vesicles (sEVs) secreted by cells that potently affect cell–cell communication[11–13]. The heterogeneity of cargo expression in exosomes underlies the diverse functions of exosomes[13]. Elucidating the mechanisms of cargo sorting to exosomes is key to understanding the heterogeneity of exosomes and their functions. The endosomal sorting complexes required for transport (ESCRT) machinery play an important role in exosome biogenesis[12,14]. HRS (also known as HGS, hepatocyte growth factor-regulated tyrosine kinase substrate), is a key component of ESCRT as it mediates the initial cargo recognition and sorting into multivesicular endosomes (MVEs), which are then delivered to the plasma membrane for exosome secretion[14]. Tumor cells secrete exosomes that carry PD-L1, a key immune checkpoint protein[15–22]. How PD-L1 loading to exosomes is regulated in tumor cells is unknown.

Here, we report that, following phosphorylation by an extracellular signal-regulated kinase (ERK), HRS spatially excludes the infiltration of CD8[+] T cells into melanoma tumor tissues. Mechanistically, phosphorylated HRS strongly interacts with PD-L1, and selectively promotes PD-L1 loading to the exosomes, thereby blocking CD8[+] T-cell infiltration. In various murine models, the expression of constitutively phosphorylated HRS leads to resistance to anti-PD-1 treatment, whereas inhibition of HRS phosphorylation enhances the therapeutic efficacy of PD-1 blockade. Our study reveals a mechanism by which oncogenic signaling regulates anti-tumor immunity through PD-L1 loading to the exosomes, and suggests inhibiting HRS phosphorylation as a potential strategy to enhance ICB-based therapies.

## Results

### Phosphorylation of HRS by ERK restricts infiltration of CD8[+] T cells into tumors.

Cancer cells secrete exosomes that influence the tumor microenvironment and immune system[11,23,24]. As HRS plays an important role in exosome biogenesis, we examined the potential phosphorylation of HRS by oncogenic kinases in cancer cells. Using mass-spectrometry, we analyzed HRS purified from the metastatic melanoma cell line, WM9. We identified a phospho-peptide "KS*PTPSAPVPLTEPAAQPGEG", in which Serine 345 ("S345") was phosphorylated (Fig. 1a and Supplementary Fig. 1a). S345 matches the ERK consensus phosphorylation site. To verify the phosphorylation of HRS by ERK on S345, we mutated this serine residue to alanine ("S345A"). This point mutation abolished the ability of HRS to be recognized by the anti-ERK phospho-substrate antibody (Fig. 1b). As a control, mutating the adjacent threonine residue (T347), which also matches the ERK consensus phosphorylation sequence, did not abolish HRS phosphorylation (Fig. 1b). To test whether ERK phosphorylates HRS in vitro, we expressed Flag-tagged HRS in HEK293T cells. We then purified Flag-HRS from cell lysates, and treated it with recombinant constitutively activated ERK ("CA-ERK") or kinase-dead ERK ("KD-ERK") purified from bacteria. CA-ERK, but not KD-ERK, phosphorylated HRS, suggesting ERK phosphorylates HRS directly (Fig. 1c). In cells, HRS phosphorylation is increased in response to epidermal growth factor (EGF) treatment, which is known to induce ERK activation. Treatment

of cells with the ERK inhibitor SCH772984 blocked the HRS phosphorylation (Fig. 1d).

Based on HRS amino-acid sequence, we generated an antibody using HRS phospho-peptide so that it can specifically recognize HRS that is phosphorylated at S345 ("p-HRS[S345]"). This antibody detected phosphorylated HRS in cells treated with EGF, and the detection was abolished after treatment with SCH772984 (Supplementary Fig. 1b). We also tested the antibody on tumor tissues by immunohistochemistry (IHC). Staining of tumor xenografts by this antibody showed that the level of p-HRS was much lower in the group of tumor tissues treated with BVD-523, an ERK inhibitor used in clinical trials (Supplementary Fig. 1c).

Using this antibody, the distribution of p-HRS[S345] and CD8[+] T cells in tumor tissues was examined on tissue microarrays (TMA) that contain samples from melanoma patients (Supplementary Fig. 2a). There were lower levels of CD8[+] tumor-infiltrating lymphocytes (TILs) in tumors with high levels of p-HRS[S345] compared to tumors with low p-HRS[S345] (Fig. 1e, f). The numbers of CD8[+] TILs were inversely correlated with the level of p-HRS[S345] in primary melanomas ($R = -0.4060$; $P < 0.001$) (Fig. 1g). As a control, there was no correlation of CD8[+] TILs with the levels of total HRS protein (Fig. 1h, i). A similar distribution pattern was found in melanoma metastasized to lymph nodes (Supplementary Fig. 2b–f).

In individual tumors, p-HRS[S345] was heterogeneously expressed (Fig. 1j). CD8[+] TILs were greatly diminished in the areas with high p-HRS[S345] (Fig. 1j). In order to compare IHC scoring of CD8[+] TILs in regions where tumor cells had high versus low pHRS, we examined 20 tumors from melanoma patients using computer-aided imaging analysis (see METHODS for details). A significantly lower level of CD8[+] TILs was observed in regions with tumor cells expressing high p-HRS[S345] compared to regions with low p-HRS[S345] (Supplementary Fig. 2g, h). At the tumor-stroma border, fewer CD8[+] T cells migrated into tumor tissues where there were high levels of p-HRS[S345] (Fig. 1k, l). Together, these results suggest that HRS[S345] phosphorylation is associated with spatial restriction of CD8[+] T cells in melanoma.

### HRS phosphorylation suppresses TILs and induces resistance to anti-PD-1 treatment.

CD8[+] T-cell infiltration is associated with anti-tumor immunity. To investigate whether HRS phosphorylation in tumor cells is involved in immunosuppression, we established stable YUMMER1.7 cell lines that express wild-type ("HRS[WT]"), phospho-deficient ("HRS[S345A]"), and phospho-mimetic mutant HRS ("HRS[S345D]") (Supplementary Fig. 3a). Interestingly, YUMMER1.7 tumors expressing HRS[S345D] grew significantly faster than tumors expressing HRS[WT] and HRS[S345A] in C57BL/6 mice but not in Rag2[−/−] immune-deficient mice (Supplementary Fig. 3b, c), suggesting the involvement of the immune system. Flow cytometry analysis showed that the numbers of PD-1[+] CD8[+] TILs were significantly lower in tumors expressing HRS[S345D] compared to those expressing HRS[WT] or HRS[S345A] (Fig. 2a), consistent with the negative correlation of p-HRS[S345] and CD8[+] TILs in melanoma patients shown by IHC (Fig. 1e–g). Furthermore, the amount of CD8[+] TILs expressing Ki-67 and Granzyme B was lower in tumors expressing HRS[WT] or HRS[S345D] compared to those expressing HRS[S345A] (Fig. 2b). These results suggest that HRS phosphorylation inhibits the infiltration of functional CD8[+] T cells.

Next, we examined whether HRS phosphorylation affects PD-1 blockade treatment on YUMMER1.7 tumors expressing different HRS mutants. Anti-PD-1 antibody effectively inhibited the growth of tumors expressing HRS[S345A] (Fig. 2c). The inhibitory effect of PD-1 blockade was decreased in HRS[WT] tumors, and

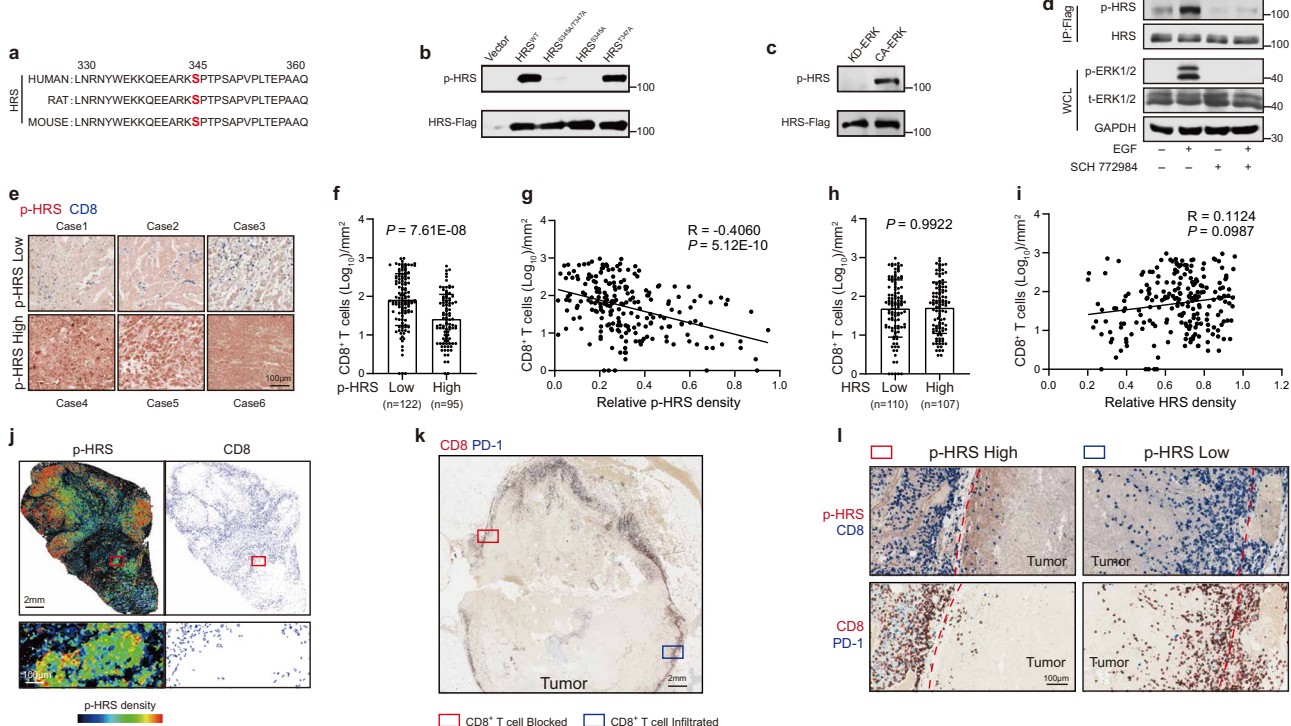

**Fig. 1 Phosphorylation of HRS by ERK restricted CD8$^+$ T cell filtration in melanoma. a** Amino-acid sequences of HRS at Serine 345 (S345) (red) across species. **b** HRS was immunoprecipitated from HEK 293 T cells expressing Flag-tagged wild-type HRS or indicated mutants. HRS phosphorylation was determined using the anti-ERK phospho-substrate antibody. **c** Purified Flag-tagged HRS was incubated with constitutively-activated ERK2 ("CA-ERK2") or kinase-dead mutant ERK2 ("KD-ERK2") in vitro and the phosphorylation was determined by anti-ERK phospho-substrate antibody. **d** Cells expressing Flag-tagged HRS were serum-starved overnight and then treated with EGF in the presence or absence of ERK inhibitor SCH772984. HRS-Flag protein was then immunoisolated from cell lysates, and phosphorylation of HRS-Flag was detected with anti-ERK phospho-substrate antibody. As a control, EGF treatment also activated ERK as shown by its phosphorylation ("pERK1/2"). Total ERK levels were also shown ("t-ERK"). GAPDH was used as a loading control. **e** Representative IHC images of melanoma tissues co-stained with antibodies against HRS phosphorylated at S345 ("p-HRS") and CD8. Tumors expressing low (upper panel) and high (lower panel) levels of pHRS are shown. Scale bar: 100 μm. **f** The numbers of CD8$^+$ TILs in melanomas from patients with different p-HRS expression levels (p-HRS-low group, $n = 122$; p-HRS-high group, $n = 95$). See METHODS for details. **g** Correlation of the numbers of CD8$^+$ TILs with p-HRS levels in malignant melanoma tissues ($n = 217$). **h** The numbers of CD8$^+$ TILs in melanomas obtained from patients with different levels of HRS (HRS-low group, $n = 110$; HRS-high group, $n = 107$). **i** Correlation of the levels of CD8$^+$ TILs with HRS expression in the malignant melanoma tumors ($n = 217$). **j** Heatmap of p-HRS (left) and CD8$^+$ cells distribution (right) in melanoma tumor tissues. Red boxes highlight areas shown in zoomed inset. See METHODS for details. **k** Distribution of PD-1$^+$ (blue) and CD8$^+$ (red) cells in human melanoma. Red box highlights a representative area with CD8$^+$ T-cell blocked at the border. Blue box highlights a representative area with CD8$^+$ T-cell infiltrating into the tumor. Scale bar: 2 mm. **l** Zoomed insets of boxed areas in **k**. Red dashed lines indicate tumor boundary. Scale bar: 100 μm. The experiments were repeated two (**b**, **c**) and three (**d**) times independently with similar results obtained. Data represent mean ± s.d. Statistical analyses were performed using Spearman's correlation (**g**, **i**) and two-tailed Mann–Whitney's U test (**f**, **h**). Source data are provided as a Source Data file.

nearly abolished in HRS$^{S345D}$ tumors (Fig. 2c). These results suggest that immunosuppression induced by HRS phosphorylation hindered the effect of PD-1 blockade, and raise the possibility that inhibition of HRS phosphorylation would enhance the therapeutic effect of PD-1 blockade. To test this possibility, we combined anti-PD-1 antibody and ERK inhibitor BVD-523 in B16F10 tumors, which, unlike YUMMER1.7 tumors, are known to be refractory to PD-1 blockade[25,26]. B16F10 tumors harboring different HRS variants did not show significant differences in growth in C57BL/6 mice (Supplementary Fig. 3d, e). However, there was a smaller number of infiltrated CD8$^+$ T cells in tumors expressing HRS$^{WT}$ or HRS$^{S345D}$ compared to those expressing HRS$^{S345A}$ (Supplementary Fig. 3f, g). ERK inhibitor BVD-523 improved CD8$^+$ TILs in tumors expressing HRS$^{WT}$, but not in tumors expressing HRS$^{S345D}$, which mimics the phosphorylated HRS that cannot be altered by ERK inhibition (Fig. 2d–f and Supplementary Fig. 4a, b), suggesting BVD-523 functioned through HRS phosphorylation inhibition. BVD-523 treatment also led to a slight increase of T-cell infiltration in tumors

expressing HRS$^{S345A}$ (Fig. 2e), which was probably due to the inhibition of the phosphorylation of endogenous HRS. Consistent with the effect on CD8$^+$ T-cell infiltration, the combination of ERK inhibition with anti-PD-1 antibodies led to significant tumor regression in tumors expressing HRS$^{WT}$ and HRS$^{S345A}$, not in tumors expressing HRS$^{S345D}$ (Fig. 2d–f and Supplementary Fig. 4a, b). Taken together, these results strongly suggest that HRS phosphorylation suppresses the infiltration of CD8$^+$ T cells and contributes to resistance to PD-1 blockade; inhibiting HRS phosphorylation enhances the therapeutic effects of PD-1 blockade.

### HRS phosphorylation leads to selective enrichment of PD-L1 in exosomes

Next, we investigated the mechanism by which HRS phosphorylation regulates T-cell infiltration. In PD-L1 knocked out ("PD-L1-KO") B16F10 tumors, the expression of HRS$^{S345D}$ failed to suppress CD8$^+$ TILs (Supplementary Fig. 4c–f), suggesting an important role of PD-L1 in HRS phosphorylation-induced immunosuppression. HRS is a key component of the

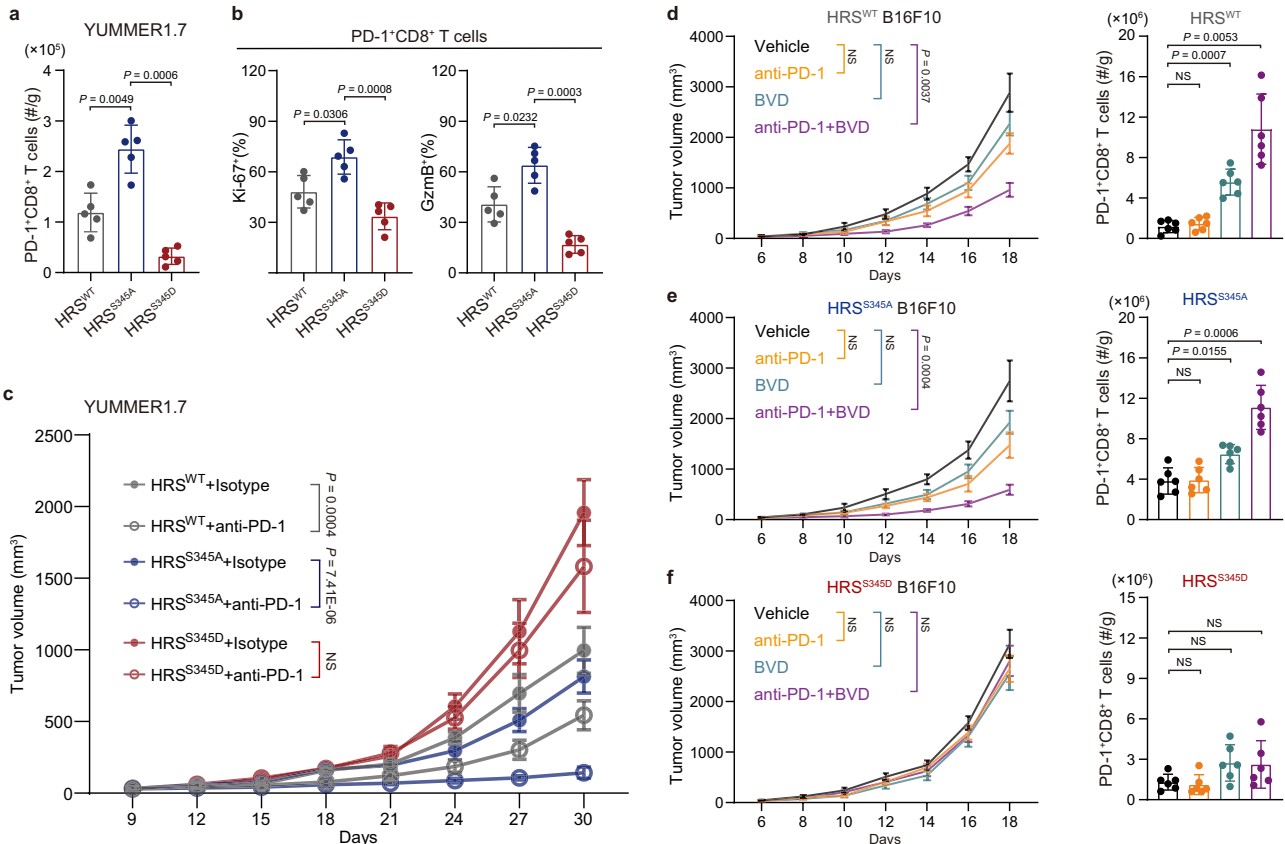

**Fig. 2 HRS<sup>S345</sup> phosphorylation blocks CD8<sup>+</sup> T cell infiltration and reduces the efficiency of anti-PD-1 treatment in mice. a** Bar graphs showing the numbers of intratumor PD-1+CD8+ TILs (normalized to tumor weights) isolated from the YUMMER1.7 tumors expressing different HRS variants ($n = 5$). **b** Percentages of Ki-67+ (left) and Granzyme B+ (right) cells of PD-1+ CD8+ T cells in YUMMER1.7 tumors ($n = 5$). **c** Growth of different HRS-expressing YUMMER1.7 tumors in C57BL/6 mice treated with anti-PD-1 or isotype control antibodies ($n = 5$). **d–f** Growth of B16F10 tumors expressing HRS<sup>WT</sup> (**d**, left), HRS<sup>S345A</sup> (**e**, left), or HRS<sup>S345D</sup> (**f**, left) treated with vehicle, anti-PD-1, BVD-523 (BVD), or anti-PD-1 plus BVD-523 as indicated. The numbers of PD-1+CD8+ TILs (normalized to tumor weights) were shown in right ($n = 6$). The experiments were repeated three times independently with similar results obtained (**a–f**). Data represent mean ± s.d. Statistical analyses were performed using one-way ANOVA (**a**, **b**, **d** right, **e** right, **f** right) or two-way ANOVA (**c**, **d** left, **e** left, **f** left). Tukey's test was used following ANOVA. Source data are provided as a Source Data file.

ESCRT complex that mediates the sorting of PD-L1 into MVEs, from which PD-L1 can either be routed to lysosomes for degradation, which is critical to PD-L1 surface expression[27,28], or to the extracellular space as exosomal PD-L1, which contributes to tumor immunosuppression[15]. We found that cell surface PD-L1 in human WM9 cells or mouse B16F10 cells were not changed by the expression of the HRS mutants (Supplementary Fig. 5a–d). We next examined the levels of PD-L1 on small extracellular vesicles ("sEVs", diameter <200 nm), which mostly consist of exosomes derived from these cells. There was no difference in the total number of sEVs secreted by cells expressing different HRS mutants, as examined by nanoparticle tracking analysis (Supplementary Fig. 5e, f). Next, we analyzed the protein compositions of the sEVs. sEVs derived from WM9 cells expressing different HRS mutants were isolated by differential centrifugation. The proteins on the sEVs were analyzed by mass-spectrometry (Fig. 3a, b, and Supplementary Fig. 5g–i). The proteins exhibiting most significant differences among the HRS mutants are shown in Fig. 3c and Supplementary Fig. 5g, j. PD-L1 was identified in the group of membrane proteins that were most significantly enriched in the sEVs derived from HRS<sup>S345D</sup> compared to those from HRS<sup>S345A</sup> cells (Fig. 3c). On the other hand, no significant upregulation was found for the well-known exosome proteins such as GPCRs, RTKs, integrins, and cadherins, as well as members of the ESCRT complex (Supplementary Fig. 5j).

The enrichment of PD-L1 in HRS<sup>S345D</sup> cell-derived sEVs was further confirmed by western blotting. Compared to sEVs from cells expressing HRS<sup>WT</sup>, the amount of PD-L1 was significantly reduced in sEVs from cells expressing HRS<sup>S345A</sup> and increased in sEVs from cells expressing HRS<sup>S345D</sup> (Fig. 3d and Supplementary Fig. 6a). We also examined the effect of the ERK inhibitor on sEVs PD-L1 expression. BVD-523 inhibited HRS phosphorylation and reduced the loading of PD-L1 onto sEVs (Fig. 3e and Supplementary Fig. 6b). In cells expressing HRS<sup>WT</sup>, BVD-523 treatment decreased the amount of PD-L1 in sEVs to a level closed to HRS<sup>S345A</sup>. In contrast, BVD-523 did not change the level of PD-L1 in sEVs derived from HRS<sup>S345D</sup>-expressing cells (Supplementary Fig. 6c, d), suggesting BVD-523 regulates PD-L1 loading through HRS phosphorylation. For all of the above experiments, there was no difference in the amounts of CD63, CD9, or CD81, the commonly used exosome marker proteins, in exosomes derived from different cells.

In addition to examining exosomes from cultured cells, we also examined whether PD-L1 was enriched in tumor tissue-derived EVs ("TTDEs"). TTDEs were isolated from different mouse tumor models: human melanoma WM9 cells established in *Rag2*<sup>−/−</sup> mice, and murine melanoma B16-F10 established in C57BL/6 mice and *Rag2*<sup>−/−</sup> mice. Single-particle tracking analysis indicated that the isolated TTDEs had the correct size distribution expected for exosomes (Supplementary Fig. 7a).

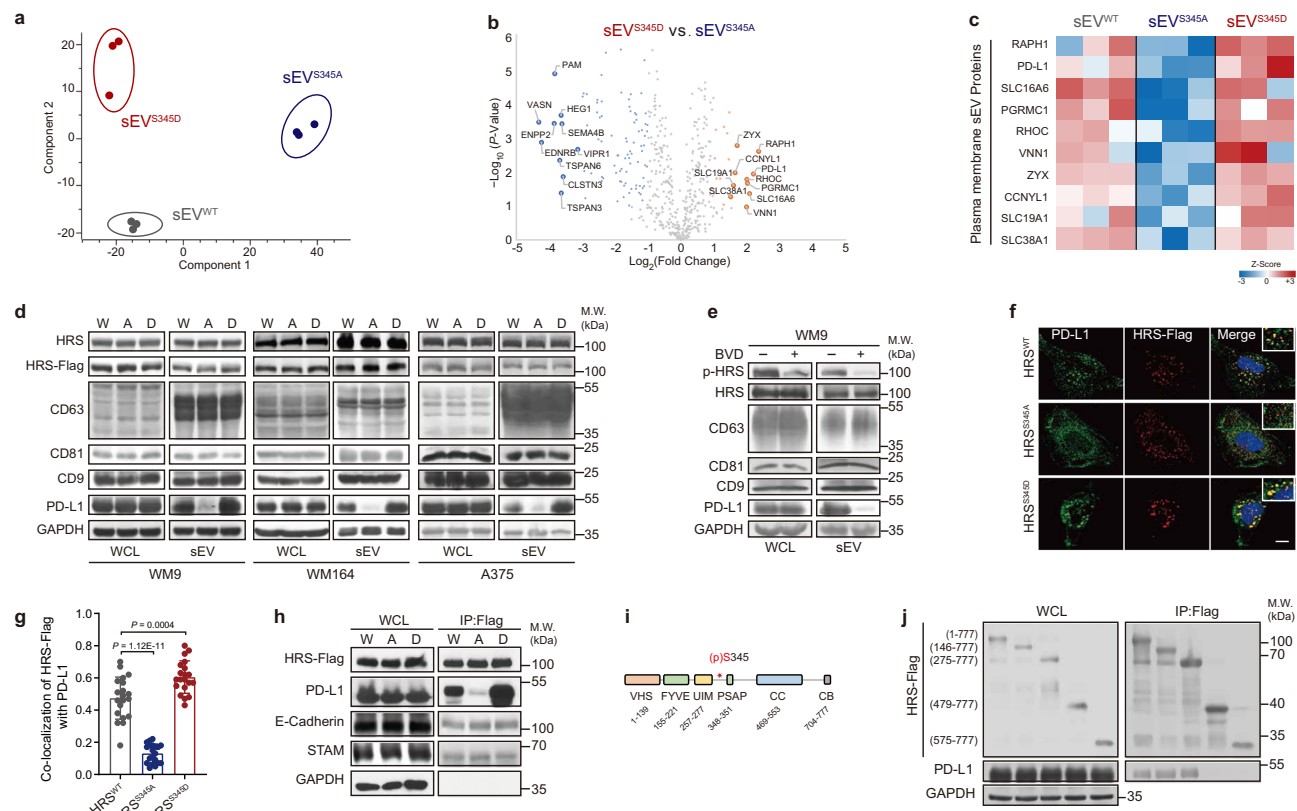

**Fig. 3 HRS$^{S345}$ phosphorylation selectively enriches PD-L1 in sEVs. a** Principal component analysis (PCA) of proteomic profiling of sEVs from WM9 cell lines expressing HRS$^{WT}$ (sEV$^{WT}$), HRS$^{S345A}$ (sEV$^{S345A}$) or HRS$^{S345D}$(sEV$^{S345D}$). **b** Volcano plot displaying membrane proteins quantified by LC-MS/MS that are changed in sEVs derived from WM9 cells expressing HRS$^{S345D}$ compared with sEVs from cells expressing HRS$^{S345A}$. Proteins were considered significantly decreased (blue) or increased (orange) based on FDR < 0.05 and fold change >2.0. The 20 proteins having the largest fold increases (≥3) or decreases (≤8) are labeled. **c** Heatmap showing z score intensity of sEV proteins (increased in **b**) among HRS$^{WT}$ and mutants. **d** The levels of HRS, PD-L1, and other exosome proteins (CD63, CD81, and CD9) in the cell lysate ("WCL") or sEVs purified from WM9, WM164, or A375 expressing HRS$^{WT}$-Flag ("W"), HRS$^{S345A}$-Flag ("A") and HRS$^{S345D}$-Flag ("D") were determined by western blot. The same amounts of proteins were loaded. GAPDH was used as a control. **e** Indicated proteins in whole-cell lysates or sEVs from WM9 cells treated with vehicle or BVD-523 (2 μM) were determined by western blot. **f** Immunofluorescence staining of PD-L1 (green) and HRS variants (red) in WM9 cells. Nuclei were stained with DAPI. Scale bar: 10 μm. **g** Percentages of co-localization of endogenous PD-L1 with different HRS variants. 20 fields of each group were measured. **h** Immunoprecipitation was performed in WM9 cells expressing Flag-tagged HRS$^{WT}$, HRS$^{S345A}$, or HRS$^{S345D}$ using an anti-Flag antibody. Proteins in whole-cell lysate or precipitates (IP: Flag) were determined by western blot. GAPDH was used as loading control. **i** Diagram of domains of HRS. HRS binds the ubiquitinated cargo via the ubiquitin-interacting motif (UIM). The phosphorylation site S345 is located after the UIM domain. **j** Immunoprecipitation from cells expressing PD-L1 and Flag-tagged full-length or fragments of HRS using an anti-Flag antibody. Indicated proteins in whole-cell lysate or precipitates were determined by western blotting. GAPDH was used as loading control for cell lysates. Left panel shows the levels of proteins in cell lysates; the right panel shows the immunoprecipitated HRS fragments and bound PD-L1. Data are representative of three independent experiments (**d**, **e**, **h**, **j**) or are pooled from two (**a**, **b**, **c**) or four experiments (**f**, **g**). Data represent mean ± s.d. Statistical analyses were performed using permutation-based FDR (**b**) or one-way ANOVA (**g**). Tukey's test was used following ANOVA. Source data are provided as a Source Data file.

TTDEs from tumors expressing HRS$^{S345A}$ ("TTDE$^{S345A}$") had a lower level of PD-L1 compared to TTDEs from tumors expressing HRS$^{WT}$ (TTDE$^{WT}$) (Supplementary Fig. 7c–e). BVD-523 reduced the levels of PD-L1 in TTDEs from normal tumors or tumors expressing HRS$^{WT}$ but not tumors expressing HRS$^{S345D}$ (Supplementary Fig. 7b, c, and e). These results suggest that HRS phosphorylation by ERK specifically enriches PD-L1 in exosomes.

PD-L1 is sorted into exosomes by HRS, and our previous study showed that PD-L1 co-localized with HRS on MVEs[15]. Compared to the wild-type HRS, PD-L1 showed significantly decreased co-localization with HRS$^{S345A}$ and increased co-localization with HRS$^{S345D}$ (Fig. 3f, g). Furthermore, immuno-precipitation studies demonstrated that PD-L1 interacted strongly with HRS$^{S345D}$, while the interaction was diminished by the S345A mutation (Fig. 3h). Neither mutation affected the interaction of HRS with STAM, a known partner of HRS in the ESCRT, or E-Cadherin, which was previously shown to interact with HRS (Fig. 3h)[29,30], consistent with the proteomics data (Supplementary Fig. 5j). The results suggest that ERK-mediated phosphorylation of HRS promotes the recruitment of PD-L1 to the MVEs for exosome secretion.

HRS is thought to bind to ubiquitinated cargos such as E-Cadherin through its ubiquitin-interacting motif (UIM)[30–32]. Ubiquitin-independent cargo binding by HRS was also reported but the nature of the interaction is unclear[33]. Our domain mapping experiments showed that HRS (a.a. 275–777) lacking UIM retained PD-L1 binding, whereas further deletion of a.a. 276–478 abolished the interaction (Fig. 3i, j). Given that S345 is located within the region for PD-L1 binding, the data suggest that S345 phosphorylation specifically modulates ubiquitin-independent sorting.

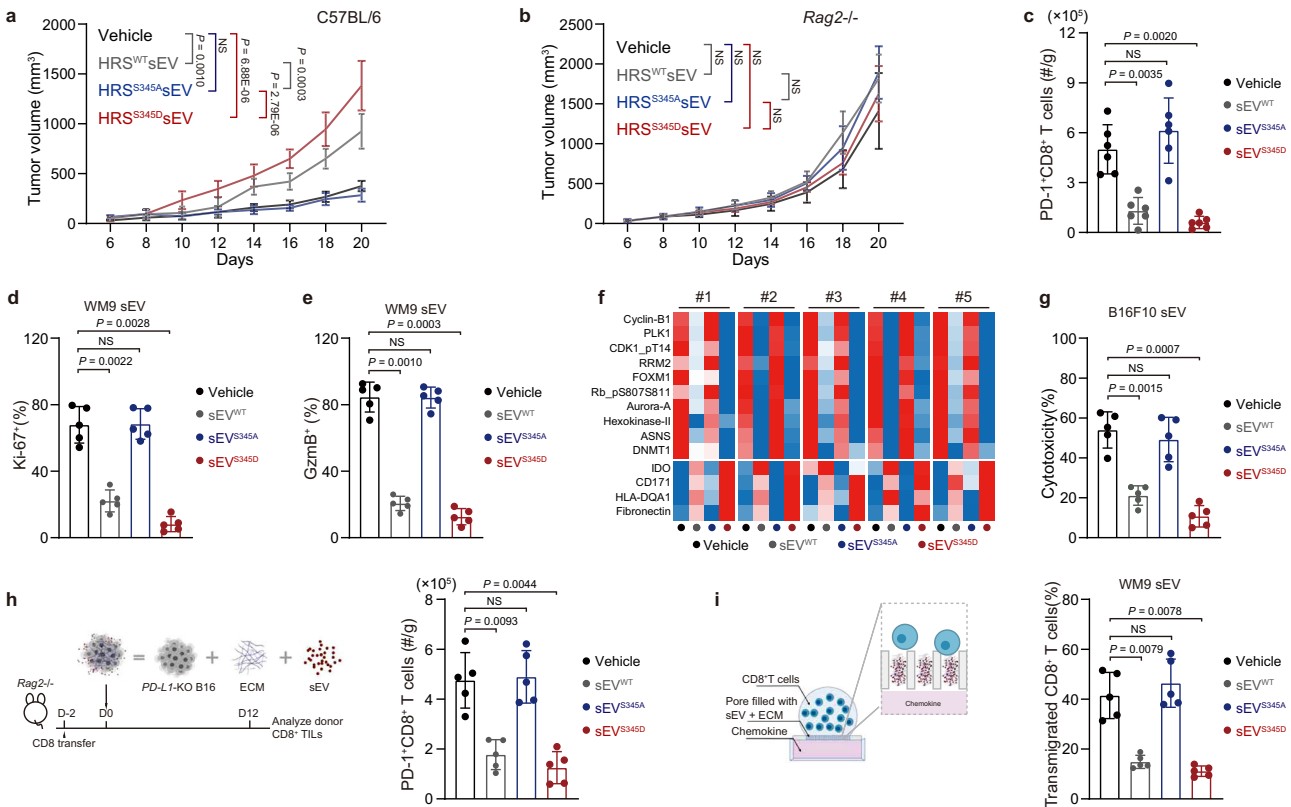

**Fig. 4 HRS phosphorylation enhances the suppressive effect of sEV on CD8$^+$ T cells. a–b** Growth of *PD-L1*-KO B16F10 tumors in C57BL/6 and *Rag2$^{-/-}$* mice with indicated sEV treatments ($n = 6$ for each group). **c** Scatter-bar graphs indicating the number of intratumor PD-1$^+$CD8$^+$ TILs (normalized to tumor weights) in B16F10 tumors expressing HRS mutants in (**a**). **d, e** Percentages of Ki-67$^+$ and Granzyme B$^+$ cells in human CD8$^+$ T cells treated with indicated sEVs ($n = 5$). **f** Heatmap of RPPA data showing the significantly changed proteins in peripheral CD8$^+$ cells with indicated sEVs treatment. Proteins considered significantly changed between vehicle control and sEV$^{WT}$ treatment group are described in Supplementary Fig. 9d. **g** Cytotoxicity elicited by mouse splenocytes with or without exposure to indicated B16F10 cell-derived sEVs ($n = 5$). See *METHODS* for details. **h** Schema for sEV co-xenograft system (left) (See details in *METHODS*). CD8$^+$ T cells were transferred into *Rag2$^{-/-}$* mice 2 days before tumor injection. *PD-L1*-KO B16F10 cells were mixed with Matrigel (ECM) and sEVs from B16F10 cells and subcutaneously inject into the mice. The number of PD-1$^+$CD8$^+$ T cells in the co-grafted tumors was determined on Day 12 ($n = 5$) (right). **i** Schema for the chemotaxis assay to access CD8$^+$ T cells migration with sEVs and fibronectin (ECM) (left) (See the details in *METHODS*). Percentages of transmigrated human CD8$^+$ T cells stimulated with CXCL9 (100 ng/mL) in chemotaxis assay. 3-μm size pores were blocked by fibronectin with vehicle or indicated WM9 cell-derived sEVs ($n = 5$) (right). The experiments were repeated three (**a–c**, **h**) and five (**d, e, g, i**) times independently with similar results obtained. Data represent mean ± s.d. Statistical analyses were performed using one-way ANOVA (**c–e, g–i**) or two-way ANOVA (**a, b**). Tukey's test was used following ANOVA. Source data are provided as a Source Data file.

### sEVs derived from cells with HRS$^{S345D}$ effectively suppress CD8$^+$ T cell infiltration.

We next examined the effect of sEVs from different HRS mutant cells on tumor progression. To specifically examine the role of these PD-L1$^+$ sEVs without the interference of PD-L1$^+$ sEV originating from the allografts, we first generated B16F10 tumor cells with their endogenous PD-L1 knocked out ("*PD-L1*-KO B16F10"). Infusion of sEV$^{WT}$ or sEV$^{S345D}$ derived from B16F10 cells expressing PD-L1 significantly promoted *PD-L1*-KO B16F10 tumor growth in C57BL/6 mice. The effect was not observed for tumors grown in *Rag2$^{-/-}$* mice (Fig. 4a, b). The infiltration of CD8$^+$ T cells to the *PD-L1*-KO B16F10 tumors was inhibited by sEV$^{WT}$ and sEV$^{S345D}$ (Fig. 4c), and TILs with Ki-67 and Granzyme B expression were reduced in these tumors (Supplementary Fig. 8a). The data are consistent with the CD8$^+$ TIL analysis in tumors expressing different HRS mutants (Fig. 2a, b; and Supplementary Fig. 3f, g).

To examine the direct effect of these sEVs on CD8$^+$ T cells, in vitro experiments were performed. Compared to PBS control or sEV$^{S345A}$, treatment with sEV$^{WT}$ and sEV$^{S345D}$ from both human and mouse melanoma cell lines significantly inhibited CD8$^+$ T cell proliferation and activation, as assessed by their expression of Ki-67 and Granzyme B (Fig. 4d, e and

Supplementary Fig. 8b–e; Supplementary Fig. 9a). Similar inhibitory effects were observed with TTDE$^{WT}$ and TTDE$^{S345D}$ from mouse tumors (Supplementary Fig. 9b, c). Further analysis of the protein expression profile of CD8$^+$ T cells treated with different sEVs by RPPA (Reverse-Phase Protein Array), an antibody-based quantitative proteomics technology, showed that sEV$^{WT}$ and sEV$^{S345D}$ significantly downregulated the expression of a cluster of proteins related to T cell proliferation and activation, such as FOXM1, Aurora A, ASNS, Cyclin-B1, while there was no such difference between sEV$^{S345A}$ and PBS (Fig. 4f and Supplementary Fig. 9d, e).

We then examined the cytotoxicity of T cells primed by B16F10 cells. Mouse splenocytes treated with sEV$^{WT}$ or sEV$^{S345D}$ showed reduced cytotoxicity against B16F10 cells compared to treatment with PBS or sEV$^{S345A}$ (Fig. 4g). We next transferred CD8$^+$ T cells that were pre-treated with different types of sEVs into *Rag2$^{-/-}$* mice bearing *PD-L1*-KO B16F10 tumors and analyzed their infiltration (Supplementary Fig. 10a). sEV$^{WT}$ and sEV$^{S345D}$ significantly inhibited CD8$^+$ T cell infiltration, whereas sEV$^{S345A}$ and vehicle did not show any effect. These data suggest that HRS phosphorylation enhances the inhibitory function of sEVs on CD8$^+$ T cells.

As shown in Fig. 1j, k, in patient melanoma tissues, CD8$^+$ T cells were excluded from regions with high levels of pHRS expression. EM imaging of sEVs isolated from tumor tissues showed that the sEVs were often associated with the ECM fibers (Supplementary Fig. 10b and 11a). These observations raised the possibility that exosomes associate with the ECM surrounding tumor cells to block T cell infiltration. Here we transferred CD8$^+$ T cells into $Rag2^{-/-}$ mice and injected $PD-L1$-KO B16F10 cells mixed with sEVs and Matrigel, which contains components of ECM and is often used to mimic ECM in vitro. We found that the infiltration of transferred CD8$^+$ T cells was inhibited by sEV$^{WT}$ and sEV$^{S345D}$ (Fig. 4h). To directly examine the function of ECM-bound sEVs on T-cell migration, we used an in vitro transmigration assay (Fig. 4i). Transwells were blocked with a mixture of sEVs and fibronectin to mimic the ECM loaded with tumor-derived sEVs. The transmigration of CD8$^+$ T cells was stimulated with CXCL9 or CXCL10. The transmigration of T cells was greatly reduced by the ECM loaded with sEV$^{WT}$/TTDE$^{WT}$ and sEV$^{S345D}$/TTDE$^{S345D}$, while there was no effect for sEV$^{S345A}$/TTDE$^{S345A}$ as compared to vehicle control (Fig. 4i and Supplementary Fig. 10c–e and Supplementary Fig. 11b–e). These data suggest that, in tumor ECM, exosomes derived from tumors with HRS phosphorylation directly inhibit CD8$^+$ T cell infiltration.

Since PD-L1 was enriched in the secreted exosomes but not altered on the cell surface as a result of HRS phosphorylation (Supplementary Fig. 5a–d), we examined the role of sEV PD-L1 in pHRS-induced CD8$^+$ T cell suppression (Supplementary Fig. 4d–f). sEV$^{WT}$ and sEV$^{S345D}$ derived from $PD-L1$-KO B16F10 cells did not show any inhibitory effect on the cytotoxicity of CD8$^+$ T cells (Fig. 5a). For sEV$^{WT}$/ TTDE$^{WT}$ and sEV$^{S345D}$/ TTDE$^{S345D}$ derived from cells or tumors expressing PD-L1, pretreatment with anti-PD-L1 blocking antibodies attenuated their inhibitory effects on CD8$^+$ T cells (Fig. 5b and Supplementary Fig. 12a–d). Mouse sEV infusion experiments showed that PD-L1 blockade on sEVs attenuated their tumor growth-promoting effect and inhibition of CD8$^+$ TILs by sEV$^{WT}$ and sEV$^{S345D}$ (Fig. 5c, d; Supplementary Fig. 12e). To specifically analyze the role of PD-L1 on sEVs' inhibitory effect in vivo, we set up an assay taking advantage of the CD45.1$^+$ and CD45.2$^+$ sub-populations of CD8$^+$ T cells. The two homologs of CD45, CD45.1, and CD45.2, have the same functions but contain unique epitopes that can be recognized by different monoclonal antibodies[34,35]. Isolated CD45.1$^+$ CD8$^+$ T cells were pre-incubated with different sEVs, and CD45.2$^+$ CD8$^+$ T cells were pre-incubated with the same set of sEVs pre-treated with PD-L1-blocking antibodies. The CD45.1$^+$ CD8$^+$ T cells and CD45.2$^+$ CD8$^+$ T cells were then co-transferred into the same $Rag2^{-/-}$ mice bearing $PD-L1$-KO B16F10 tumor at 1:1 ratio, so that they function in the same tumor microenvironment. Tumor infiltration of these two sub-groups of T cells was examined. The amount of tumor-infiltrated CD45.1$^+$ CD8$^+$ T cells pre-incubated with sEV$^{WT}$ or sEV$^{S345D}$ was significantly lower compared to those treated with sEV$^{S345A}$, while there was no difference among the infiltrated CD45.2$^+$ CD8$^+$ T cells treated the same set of EVs, but having their PD-L1 blocked (Fig. 5e). The direct effect of PD-L1 on T cell infiltration was also examined. sEVs isolated from the $PD-L1$-KO B16F10 expressing HRS$^{WT}$ or HRS$^{S345D}$ failed to inhibit T cell transmigration in vitro (Supplementary Fig. 13a) or block T cell infiltration in vivo (Supplementary Fig. 13b), compared to the vehicle control or sEVs isolated from the $PD-L1$-KO B16F10 cells expressing HRS$^{S345A}$. Taken together, the results demonstrate that PD-L1 is required for the inhibitory effect on CD8$^+$ T cells infiltration from sEVs induced by HRS phosphorylation.

As the enrichment of PD-L1 on the sEVs can be blocked by ERK inhibitor (Fig. 3e), we examined whether BVD-523 treatment of tumors can modulate the inhibitory effect of sEVs. sEVs from cells treated with BVD-523 lost their inhibitory effect on T cell activation, migration, and cytotoxicity (Fig. 5f, g and Supplementary Fig. 13c–e). The inhibitory effect of sEV$^{S345D}$, however, was not blocked by BVD-523 (Fig. 5f, g and Supplementary Fig. 13c–e), consistent with the result that ERK inhibition did not reduce the exosomal enrichment of PD-L1 in cells expressing HRS$^{S345D}$ that mimics the phosphorylated HRS (Supplementary Fig. 6d, e). We also performed a reconstitution experiment using $Rag2^{-/-}$ mice with transferred CD8$^+$ T cells as described above (Fig. 4h). sEV$^{WT}$ from cells treated with BVD-523 show decreased inhibitory effect (Fig. 5h). In contrast, there was no difference between control and BVD-523 treatment for sEVs derived from HRS$^{S345D}$ expressing cells. These results further support the model that HRS phosphorylation modulates the inhibitory effect of exosomes on immunosuppression.

## Discussion

Infiltration of functional cytolytic T cells into tumors is closely associated with patient response to ICB[4,36–39]. To develop effective new strategies to improve patient response, it is crucial to understand the molecular mechanisms that regulate T cell infiltration into or exclusion from tumors[40,41]. Here we report the identification of the phosphorylation of HRS in response to oncogenic signaling. Tumor expression of pHRS$^{S345}$ is inversely correlated with CD8$^+$ T cell infiltration in melanoma tissues. In individual tumors, CD8$^+$ TILs were excluded from the areas with high levels of p-HRS$^{S345}$. Interestingly, expression of HRS$^{S345D}$, which mimics constitutively phosphorylated HRS, blocked CD8$^+$ T cell infiltration, and led to resistance to PD-1 blockade treatment. We also found that inhibition of HRS$^{S345}$ phosphorylation sensitized tumors to anti-PD-1 antibody treatment. Our study suggests that developing small molecule inhibitors targeting phospho-HRS$^{S345}$ may potentially minimize the toxicity often observed in the combination therapy while achieving improved efficacy of PD-1 blockade.

HRS mediates protein sorting to the exosomes. Our biochemistry and flow cytometry analyses indicate that HRS phosphorylation enriches PD-L1 to the exosomes without affecting PD-L1 expression on tumor cell surface. sEVs derived from phospho-mimetic mutant HRS$^{S345D}$ contained higher levels of PD-L1, whereas those derived from phospho-deficient mutant HRS$^{S345A}$ contained lower levels of PD-L1. Since the HRS$^{S345A}$ mutant was expressed in cells with endogenous HRS, the observed effect probably resulted from a dominant-negative effect of the mutant. In the EV field, how cargo sorting is regulated is a major question. Ubiquitination controls cargo sorting into exosomes. Through the ubiquitin-interacting motif (UIM), HRS bind to and internalize ubiquitinylated cargo to intraluminal vesicles (ILVs) to form MVEs, which eventually fuse with the plasma membrane to release these ILVs as exosomes[11,14]. Ubiquitin-independent cargo binding by HRS was also reported, although the molecular nature of this mode of sorting is unknown[33]. Our data show that HRS phosphorylation by ERK led to selective enrichment of cargo proteins including PD-L1 to exosomes. The interaction of HRS with PD-L1 is not mediated by its UIM, but through a region containing a.a. 276–478. S345 is located within the binding region. Phosphorylation of S345 increased HRS binding to PD-L1, but did not affect the conventional ubiquitin-dependent cargo binding (e.g., E-Cadherin) or the interaction with the other ESCRT-0 protein, STAM. The ubiquitin-independent binding and its regulation by HRS

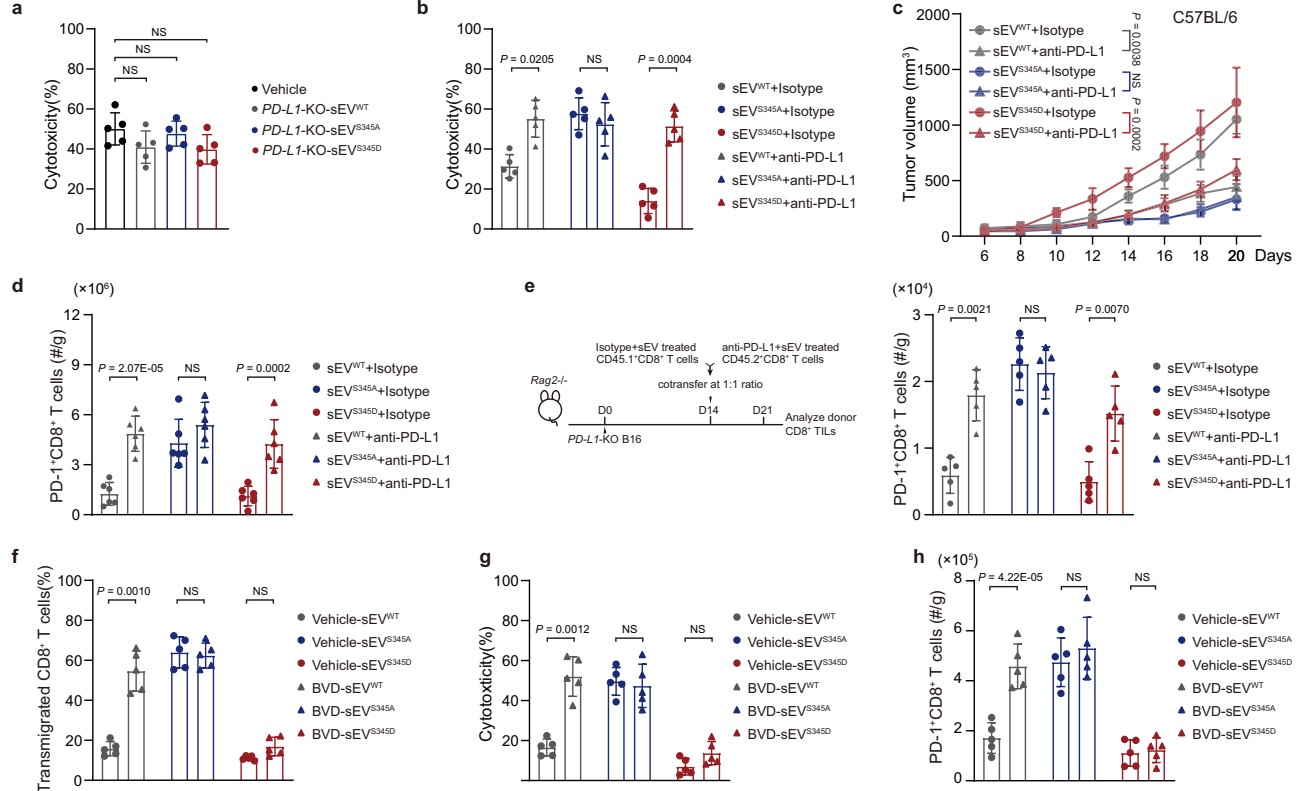

**Fig. 5 PD-L1 enriched in exosomes mediated CD8$^+$ T cell suppression induced by HRS phosphorylation. a** Cytotoxicity elicited by primed mouse splenocytes treated with or without sEVs derived from *PD-L1*-KO B16F10 cells expressing indicated HRS mutants (*n* = 5). **b** Cytotoxicity elicited by primed splenocytes treated with or without sEVs derived from B16F10 cells expressing indicated HRS mutants. sEVs were pretreated with anti-PD-L1 blocking antibody or isotype antibody controls (*n* = 5). **c** The growth of *PD-L1*-KO B16F10 tumors in C57BL/6 treated with indicated sEVs from B16F10 cells expressing various HRS mutants. The sEVs were preincubated with isotype control or anti-PD-L1 antibodies (*n* = 6). **d** The number of intratumor PD-1$^+$CD8$^+$ T cells (normalized to tumor weight) in tumors in (**c**). **e** Experimental schema for adoptive CD45.1$^+$and CD45.2$^+$ CD8$^+$ T cells co-transfer system (left). Number of intratumor PD-1$^+$CD8$^+$ T cells (normalized to tumor weight) pre-treated with indicated sEVs blocked by isotype (CD45.1$^+$ group) control or anti-PD-L1 (CD45.2$^+$ group) (normalized to tumor weight) (right). *n* = 5. **f** Transwells were pretreated with fibronectin and sEVs from B16F10 cells with or without BVD-523. Percentages of transmigrated mouse CD8$^+$ T cells induced by CXCL9 were accessed (*n* = 5). **g** Cytotoxicity elicited by primed mouse splenocytes treated with sEVs derived from B16F10 cells with or without BVD-523 treatment (*n* = 5). **h** Number of intratumor PD-1$^+$ CD8$^+$ T cells (normalized to tumor weight) in tumor co-grafted by the mixture including indicated sEVs, ECM (Matrigel) and *PD-L1*-KO B16F10 cells (*n* = 5). The experiments were repeated three (**b–h**) and five (**a**) times independently with similar results obtained. Data represent mean ± s.d. Statistical analyses were performed using one-way ANOVA (**a**) or two-way ANOVA (**b–h**). Tukey's test (**a, c**) and Sidak's (**b, d–h**) were used following ANOVA. Source data are provided as a Source Data file.

phosphorylation may offer a mechanism for the selective enrichment of PD-L1 to the exosomes in response to oncogenic signaling.

Exosomes are thought to be diffusible in body fluids and can reach the recipient T cells to change their behaviors. Exosomes were also identified in the tumor microenvironment including the ECM[42]. However, it is unclear whether the exosomes in the tumors directly bind to the ECM and whether these exosomes have special roles in tumorigenesis. A previous study suggested that ECM-attached exosomes, rather than diffusible exosomes, promote cell movement[43]. In our EM analysis of sEVs isolated from tumor tissues, we often observed a direct association of exosomes with ECM fibers, even though collagenase IV was used in the sEV isolation procedure. The observed fibers were likely the remnants of ECM digestion. While ECM disruption is necessary for the isolation of sEVs, the association of sEVs with ECM fibers would probably be more prominent if there was a lesser degree of ECM digestion. Using several parallel in vitro and in vivo models, we show that PD-L1-enriched sEVs derived from cells with HRS phosphorylation effectively suppressed the proliferation and function of CD8$^+$ T cells. We further demonstrate that PD-L1-

enriched sEVs inhibited the transmigration of CD8$^+$ T cells cross the ECM. Based on the above findings, we propose that PD-L1 enriched EVs derived from tumor cells expressing high levels of p-HRS may be distributed in the local tumor microenvironment, likely scaffolded on the ECM, to block the infiltration of CD8$^+$ T cells (Supplementary Fig. 14).

In melanoma patients, the combination of *Braf$^{V600E}$* and/or MEK inhibitors with anti-PD(L)-1 antibodies has demonstrated improved efficacy in treating patients with metastatic melanoma, and MAPK pathway inhibitors (e.g., MEK inhibitor) were shown to help reverse CD8$^+$ T-cell exclusion when combined with anti-PD-1 treatment[44–49]. Here, animal experiments were performed to investigate whether HRS phosphorylation by ERK affects anti-PD-1 antibody treatment. We found that the ERK inhibitor reduced the level of exosomal PD-L1 by inhibiting HRS phosphorylation, and sensitized B16F10 tumors to anti-PD-1 antibodies. Importantly, the expression of HRS$^{S345D}$, which represents the constitutively phosphorylated HRS, blocked CD8$^+$ T cell infiltration and abolished the response to PD-1 blockade, strongly suggesting a crucial role of HRS phosphorylation in conferring resistance to anti-PD-1 treatment. Developing small

molecules targeting phosphorylated HRS may potentially achieve improved efficacy of PD-1 blockade while minimizing the toxicity often observed in the combination therapy.

Recently, several studies have examined the levels of exosomal PD-L1 in the blood of cancer patients, and suggest the potential use of exosomal PD-L1 as an indicator or predictor of patient response to ICB-based therapies[15,16,21,50–52]. Our finding of the selective enrichment of PD-L1 by HRS phosphorylation may provide a mechanism underlying the different levels of PD-L1 in circulating exosomes observed in patients. Future studies will be needed to investigate whether pHRS$^{S345}$ IHC, combined with an assay of exosomal PD-L1 and genome mutational profiling, will provide important diagnostic information that helps select patients likely to benefit from ICB, a key opportunity given the toxicity of these therapies.

## Methods

**Cell culture and plasmids.** Human melanoma cell lines WM9, WM164, and A375 were cultured in RPMI 1640 medium (Invitrogen) supplemented with 10% (v/v) fetal bovine serum (FBS, Gibco). B16F10, YUMM1.7 and YUMMER1.7 cells were cultured in DMEM (Sigma) supplemented with 10% (v/v) FBS. All cultures were maintained in a humidified 5% $CO_2$ incubator (Thermo) at 37 °C. HRS wild-type, S345A, and S345D mutants were constructed in pCMV 3 × Flag and pBabe vectors. To establish cell lines stable expressing HRS variants, 3 × Flag-tagged HRS WT/S345A/S345D in pBabe vector were transfected into melanoma cell lines (WM9, WM164, A375, YUMM1.7, YUMMER1.7, B16F10, *PD-L1*-KO B16F10) and selected by puromycin.

**In vitro kinase assays.** Flag-tagged HRS was expressed and purified from HEK293T with anti-Flag M2 resin (Sigma). HRS-Flag proteins were incubated with His×6-ERK2-CA (constitutively active) or His×6-ERK2-KD (kinase dead) purified from *E. coli* in the kinase buffer (20 mM HEPES, pH 7.5, 100 mM KCl, 5 mM $MgCl_2$, 1 mM DTT, 1 mM NaF, and 1 mM PMSF) in the presence of ATP for 5 min at 30 °C.

**Immunoprecipitation.** Indicated cells were lysed in the NP-40 extraction buffer (25 mM HEPES-KOH, pH 7.4, 100 mM KCl, 5 mM $MgCl_2$, 1% NP-40, 1 mM NaF, and 1 mM $NaVO_4$) containing protease inhibitors cocktail (Roche) and phosphatase inhibitors (Bimake). Lysates were then cleared by centrifugation at 12,000 × $g$ for 30 min. Equal amounts of proteins were incubated with antibodies for 2 hr at 4 °C. Immunoprecipitated proteins were collected and washed four times with extraction buffer. Proteins were subjected to SDS-PAGE and western blot analysis.

**Western blotting.** Cells were lysed in radioimmunoprecipitation assay (RIPA) buffer in the presence of protease inhibitor cocktail (Roche) and phosphatase inhibitor cocktail (Bimake). Cell lysates were then subjected to 12–15% SDS-PAGE and transferred to polyvinylidene fluoride or nitrocellulose membranes (Bio-Rad Laboratories). The membranes were blocked with 5% bovine serum albumin and probed with indicated antibodies overnight at 4 °C, followed by incubation for 1 hr at room temperature with secondary antibodies (Cell Signaling Technology) conjugated with peroxidase. CD63, CD81, CD9 were used as exosome markers. GAPDH was used as a loading control. Information about the primary antibodies was included in Supplementary Table 1.

**In-gel trypsin digestion.** Samples were resuspended in SDS-PAGE sample buffer and run for a short distance (0.5 cm) onto pre-cast NUPAGE (Thermo Fisher Scientific) 1-D SDS gels. Gels were stained with Colloidal Blue (Thermo Fisher Scientific) and the entire 0.5 cm stained gel region was excised and digested overnight using 4 ng/ml modified trypsin (Promega), as previously described[53].

**Liquid chromatography-tandem mass spectrometry (LC-MS/MS).** For identification of phosphorylation by mass spectrometry, proteins isolated by gel electrophoresis were digested with trypsin (Promega) in 100 mM $NH_4HCO_3$ (pH 8). The LC-MS/MS analysis was performed on an Easy-nLC 1000 II HPLC (Thermo Fisher Scientific) coupled to a Q Exactive HF mass spectrometer (Thermo Fisher Scientific). Peptides were loaded on a pre-column (100 μm ID, 6 cm long, packed with ODS-AQ 10 μm, 120 Å beads from; YMC) and further separated on an analytical column (75 μm ID, 15 cm long, packed with Luna C$^{18}$ 1.9 μm 100 Å resin; Welch Materials) using a linear gradient from 100% Solvent A (0.1% formic acid in $H_2O$) to 30% Solvent B (0.1% formic acid in acetonitrile), 70% Solvent A in 80 min at a flow rate of 200 nL/min. The top 20 most intense precursor ions from each full scan (resolution 60,000) were isolated for HCD MS$^2$ (resolution 30,000; normalized collision energy 30) with a dynamic exclusion time of 60 s. Precursors with a charge state of +1, +7 or above, or unassigned, were excluded.

For sEV protein identification, tryptic digests were analyzed in duplicate on a Q Exactive HF mass spectrometer (Thermo Fisher Scientific) equipped with a Nano-Acquity UPLC System (Waters) with the column heater maintained at 40 °C. Duplicate injections of each tryptic digest (technical replicates) were made using a UPLC Symmetry trap column (180 μm i.d. × 2 cm packed with 5 μm C$^{18}$ resin; Waters), and peptides were separated by reverse phase-high pressure liquid chromatography (RP-HPLC) on a BEH C$^{18}$ nanocapillary analytical column (75 μm i.d. ×25 cm, 1.7 μm particle size, Waters) at a flow rate of 200 nl/min. Solvent A was Milli-Q (Millipore) water containing 0.1% formic acid, and Solvent B was acetonitrile containing 0.1% formic acid. Peptides were eluted at 200 nl/min using an acetonitrile gradient consisting of 5–30% B over 225 min, 30–80% B over 5 min, 80% B for 10 min before returning to 5% B over 0.5 min. The column was re-equilibrated using 5% B at 300 nl/min for 5 min before injecting the next sample. To minimize carryover, a blank was run between each experimental sample by injecting water and using a 30 min gradient with the same solvents. The full MS scan was acquired in profile mode at 60,000 resolutions in a 400–2000 m/z scan range. Data-dependent MS/MS was performed on the top 20 most abundant precursor ions in every full MS scan. Unassigned, +1, and +8 or above charge ions were rejected, and peptide match was set to preferred. Precursor ions subjected to MS/MS were excluded from repeated analysis for 45 s.

**IHC.** Human melanoma tissues were obtained with informed consent according to procedures approved by the Internal Review Boards (IRB) of the Hospital of the University of Pennsylvania, Massachusetts General Hospital Cancer Center of Harvard Medical School, and the Wistar Institute. All recruited volunteers provided written informed consent. Tissue microarray was built from representative FFPE tissue blocks. Tumor areas were selected by pathologists based on hematoxylin and eosin-stained slides. Duplicate cores were punched from each case (1.0-1.5 mm in diameter). Tissue sections were subjected to antigen retrieval with Tris-EDTA buffers (Agilent-DAKO) kit at 95 °C for 15 min using TintoRetriever (BioSB). Subsequently, slides were incubated with antibodies and visualized with StayBlue (Abcam) or AEC (Vector Laboratories) chromogens. Aperio CS2 Scanner (Leica) was used for scanning at × 40 to digitize the slides. Analysis of IHC slides and TMA was performed using QuPath (version 0.2.3, https://qupath.github.io/) to quantify antigens with optical density of chromogen and generate heatmap in samples (p-HRS$^{S345}$ density cutoff = 0.3; HRS density cutoff = 0.7).

**Mice.** C57BL/6 wild-type mice expressing CD45.1 (Strain number: 002014) or CD45.2 (Strain number: 000664) and *Rag2*$^{-/-}$ (Strain number: 008449) mice were purchased from Jackson Laboratories. Mice were housed in the University of Pennsylvania Animal Care Facilities under specific pathogen-free (SPF) conditions at 23 ± 2 °C ambient temperature with 40% humidity and a 12 hr light/dark cycle (7 am on and 7 pm off). Experimental and control mice were bred separately. Both males and females between the ages of 6 and 8 weeks were used in the study. Mice were euthanized via cervical dislocation. All animal procedures were pre-approved by the Institutional Animal Care and Use Committee (IACUC) of the University of Pennsylvania, and all experiments conform to the relevant regulatory standards.

**Tumor models and treatments.** For establishing melanoma xenograft model in C57BL/6 wild-type or *Rag2*$^{-/-}$ mice, WM9 cells (5 × 10$^6$ cells), YUMMER1.7 (2 × 10$^6$ cells) and B16F10 (0.5 × 10$^6$ cells) were injected subcutaneously into each mouse. Tumors were measured every other day using a digital caliper and the tumor volume was calculated by the formula ((width)$^2$ × length × 0.5). Mice were euthanized and tumors harvested 18-30 days after cell inoculation, or the longest dimension of the tumors reached 2.0 cm, as required by IACUC. For the anti-PD-1 antibody treatment, each mouse received intraperitoneal injections of 80 μg anti-mouse PD-1 (BioXcell) or Armenian hamster IgG control (BioXcell), once every 3 days, as previously described[25], starting from the third day after cell inoculation. For the BVD-523 (MedChemExpress) treatment animals were randomized into indicated groups to receive a 0.2 ml suspension containing either vehicle, BVD-523 (50 mg/kg twice daily, b.i.d.) by oral gavage. For the sEV treatment, a total of 20 μg of B16F10 cell lines derived sEVs with or without IgG isotype or PD-L1 blocking (10 μg/ml) were i.v. injected into mice after inoculation of PD-L1 knock out B16F10 cells, once every 3 days, starting from the third day after cell inoculation.

**Adoptive transfers.** The splenocytes and lymphocytes from CD45.1 and CD45.2 mice were subjected to negative selection using EasyJet Mouse CD8$^+$ T-cell Isolation Kit (STEMCELL) and stimulated with anti-CD3 (Biolegend) and anti-CD28 (Biolegend) for 24 hr. After incubated w/o sEVs for 24 hr respectively, purified CD8$^+$ cells were mixed at 1:1 ratio and i.v. injected to *Rag2*$^{-/-}$ recipient mice bearing *PD-L1*-KO B16F10 tumor (1 × 10$^6$ cells per mouse) at day 14 post-tumor inoculation. Tumors were harvested on day 21 for CD8$^+$ cells assessment. For sEV co-xenograft experiments, Bulk CD8$^+$ cells were purified from CD45.2 mice by negative selection and injected into the *Rag2*$^{-/-}$ recipient. sEVs (200 μg per mouse) and Matrigel (Corning) were pre-incubated for 24 hr at 1:2 ratio (v/v). *PD-L1*-KO B16F10 cells (1 × 10$^6$ cells per mouse) were mixed with sEVs and Matrigel premix at 1:3 ratio inoculation (200 μL per mouse) within around 48 hr post CD8$^+$ transfer. The tumors were harvested on day 12 post inoculation for CD8$^+$ cells assessment.

**Flow cytometry**. Excised tumors were minced into small pieces and digested in DMEM (Gibco) supplemented with 0.5 mg/ml collagenase type IV (Gibco) and 0.1 mg/ml DNase I (Sigma) for 30 min at 37 °C. Digested cell suspension was then processed through a 70 μm cell-strainer and rinsed with DMEM. After red blood cell lysis, Tumor-Infiltrating Leukocytes (TILs) were isolated by Percoll Gradient Centrifugation. Cells were then permeabilized in 0.1% Triton X-100, stained with antibodies in flow cytometry staining buffer, and fixed in 1% paraformaldehyde. Stained cells were then analyzed on an LSR II flow cytometer (BD Biosciences). Data were analyzed with the FlowJo software (Version 10.6.2, Treestar Inc.).

**Purification of sEVs**. To collect sEVs from cultured melanoma cells, conditioned media were harvested and sEVs were isolated by differential centrifugation as previously described[15]. Briefly, the conditioned media were centrifuged at 3000 × g for 30 mins (Beckman Coulter, Allegra X-14R) to remove apoptotic bodies and debris, followed by 16,500 × g centrifugation for 40 mins to remove microvesicles. The supernatant was further centrifuged at 120,000 × g for 2 hr to collect the sEVs. The sEVs were characterized by western blotting following the MISEV 2018 guidelines[54].

Tumor-derived sEVs were obtained as described[55]. Briefly, tumor tissues were excised and incubated with 0.5 mg/ml collagenase type IV (Gibco) and 0.1 mg/ml DNase I (Sigma) for 30 min at 37 °C under mild agitation (30 rpm). Suspensions were filtrated with 70 μm cell strainer placed onto a 50 mL tube and rinsed by pre-warmed PBS to favor sEV release and collection from the tissues. The remaining liquid is differentially centrifuged at 300 × g for 10 min and 2000 × g for 20 min to remove cells and tissue debris (Beckman Coulter, Allegra X-14R). The supernatant is then further centrifuged at 16,500 × g for 50 min (Beckman Coulter, J2-HS) to remove large EVs and at 120,000 × g for 2.5 hr to collect the crude fraction of small EVs (Beckman Coulter, Optima XPN-100).

The size and concentration of purified sEVs were determined using NanoSight NS300 (Malvern Instruments), which is equipped with fast video capture and particle-tracking software. For verification of tumor-derived sEVs using electron microscopy, purified sEVs suspended in PBS were dropped on formvar-carbon coated nickel grids. After staining with 2% uranyl acetate, grids were air-dried and visualized using a JEM-1011 transmission electron microscope.

**Fluorescence microscopy**. Cells were rinsed twice with PBS and were fixed for 20 min with 4% paraformaldehyde in PBS at room temperature. Then cells were rinsed twice with PBS and permeabilized with 0.1% Triton X-100 in PBS for 10 min. After rinsing twice with PBS, the cells were incubated with 1% BSA for 30 min at room temperature. Next, cells were incubated with the primary antibodies for 2 hr at room temperature or overnight at 4 °C. After rinsing three times with PBST (0.01% Triton-x), cells were incubated with secondary antibodies anti-rabbit IgG Alexa Fluor 568 and anti-mouse IgG Alexa Fluor 488 (Invitrogen). After rinsing three times with PBS, cells were stained with DAPI and mounted with ProLong Gold Antifade Mountant (Thermo Fisher). The cells were imaged using confocal microscopes and analyzed with NIS-Elements (Nikon).

**Human CD8+ T cells**. Blood samples from human healthy donors were collected by the Human Immunology Core at the University of Pennsylvania with the approval from the University of Pennsylvania Institutional Animal Care and Use Committee (IACUC). Written consent was obtained from each healthy donor before blood collection. All experiments involving blood samples from healthy donors were performed in accordance with relevant ethical regulations.

**T cell suppression assays**. To block PD-L1 on sEV surface, the purified sEVs (200 μg) were incubated with PD-L1 blocking antibodies (10 μg/ml) or IgG isotype antibodies (10 μg/ml) in 100 μl PBS, and then rinsed with 30 ml PBS and pelleted by ultracentrifugation twice to remove the non-bound free antibodies. Human peripheral CD8+ T cells (1 × 10^5 per well-96 well plate) obtained from Human Immunology Core of University of Pennsylvania or murine CD8+ T cells (1 × 10^5 per well-96 well plate) purified from splenocytes and lymphocytes using EasyJet Mouse CD8+ T-cell Isolation Kit (STEMCELL) were stimulated with anti-CD3 (2 μg/ml, Biolegend) and anti-CD28 (2 μg/ml, Biolegend) antibodies for 24 hr and then incubated with indicated WM9 cell/xenograft-derived sEVs or B16F10 cell/xenograft-derived sEVs (20 μg/ml) with or without PD-L1 blocking for 48 hr in the presence of anti-CD3 and CD28 antibodies. The treated CD8+ cells were then collected, stained, and analyzed by flow cytometry. For BVD-treatment in vitro, indicated WM9 or B16F10 cell lines were pretreated with BVD-523 at 2 μM concentration for 24 hr before sEV collection.

**Reverse-phase protein array (RPPA)**. Human CD8+ T cells were stimulated with anti-CD3 and anti-CD8 for 24 hr and treated with vehicle or sEVs derived from WM9 cell lines. After 48 hr incubation, CD8+ T cells were rinsed by RPMI 1640 and harvested with lysis buffer including protease and phosphatase inhibitor cocktail. The RPPA assay was performed by the MD Anderson Cancer Center core facility using 50 μg protein per sample. Antibodies were validated by Western blotting[56]. Methods for data analysis are included in statistical analyses.

**T cell priming and cytotoxicity**. PD-L1 knock-out (KO) B16F10 cells were irradiated with 25 Gy X-rays. Splenocytes were then cultured with (primed) or without (unprimed) irradiated PD-L1-KO B16F10 cells in the presence of IL-2 (5 IU/mL) and cocultured for 48 hr. Splenocytes cultured with concanavalin A (10 μg/mL) were used as a nonspecific T cell priming control. Priming was confirmed by IFN-γ ELISA of the supernatant. Primed splenocytes were then cocultured with sEVs (50 μg/ml) and freshly cultured PD-L1-KO B16F10 cells with target (PD-L1-KO B16F10) to effector (splenocytes) ratio (1:100) for 48 hr. The cell death associated LDH release and then percentage cytotoxicity was measured according to the manufacturer's protocol (Millipore Sigma). For splenocyte isolation, mouse spleens were crushed on the strainer and rinsed by 10% FBS DMEM to collect the cells. Then, 1 × RBC lysis buffer was used and neutralized by 10% FBS DMEM before splenocyte collection.

**Transmigration assay**. Murine CD8+ T cells stimulated with anti-CD3 and anti-CD28 for 24 hr and rested for 2 days in a complete RPMI culture medium containing 10% FBS, 1% Glutamine, 1× Pen/Strep, and 10 ng/ml IL-2 (Invitrogen). Human CD8+ T cells were stimulated with anti-CD3 and anti-CD28 for 24 hr. The chemotaxis assay was performed using 96-well ChemoTx chemotaxis system with 3 μm pore size (Neuro Probe) according to manufacturer's protocol. Briefly, the bottom wells were filled with 30 μl of migration buffer with or without 100 ng/ml murine or human CXCL9 or CXCL10 (PEPROTECH). To coat the 3 μm pore, fibronectin (30 μg/ml, Sigma) and sEVs (30 ug/ml) were mixed at 1:1 ratio and dropped onto the filter top for 24 hr at 4 °C. CD8+ T cells were applied to the top of filters rinsed with PBS. After 3-6 hr incubation at 37 °C, migrated cells collected from the bottom wells were quantified using a cell counter.

**Data and statistical analyses**. For identification of phosphorylation MS analyses, the software pFind3[57] was used to identify phosphorylated peptides by setting a variable modification of 79.966331 Da at S, T, and Y, and a neutral loss of 97.976896 Da at S and T. The mass accuracy of precursor ions and that of fragment ions were both set at 20 ppm. The results were filtered by applying a 1% FDR cutoff at the peptide level and a minimum of one spectrum per peptide. The MS^2 spectra were annotated using pLabel[58].

For sEV LC-MS/MS analyses, raw mass spectrometric data were processed using MaxQuant (Ver. 1.6.7.0)[59]. The "match between runs" option to match identifications across samples based on accurate m/z and retention times was enabled with 0.7 min match time window and 20 min alignment time window[60], and peak lists were searched against the human Uniprot database (released 10/01/2018; 196,371 entries) with a full tryptic constraint using the Andromeda search engine[61]. Precursor mass tolerance was set to 4.5 ppm in the main search, and fragment mass tolerance was set to 20 ppm. A maximum of two-missed cleavages was allowed, and minimal peptide length was set to seven amino acids. Carbamidomethyl cysteine was set as a fixed modification and methionine oxidation and N-terminus acetylation were set as variable modifications. A database of common expected contaminants including keratins and trypsin, as well as a decoy database produced by reversing the sequence of each protein, were combined with the forward database. Criteria for high confidence peptide/protein identifications included a false discovery rate (FDR) set to 1% for proteins and peptides. The relative abundance of each protein across all samples in an experiment was determined using the label-free quantitation (LFQ) option of MaxQuant[62]. Proteins that shared all identified peptides were combined into a single protein group by the MaxQuant software. In cases where all identified peptides from a protein were a subset of identified peptides from another protein, these proteins were also combined into that group. Peptides that matched multiple protein groups (i.e., "razor" peptides) were assigned to the protein group with the most unique peptides. Quantification was performed using razor plus unique peptides, including those modified by acetylation (protein N-terminal) and oxidation (Met). A minimum peptide ratio of 1 was required for protein intensity normalization, and "Fast LFQ" was enabled[62]. Protein identifications were filtered using Perseus software (Ver. 1.6.2.3; http://www.perseus-framework.org)[63] to remove decoy database reverse identifications, contaminants, and proteins identified only by site modified peptides or proteins identified by a single uniquely-mapping peptide. In addition, prior to statistical analysis, protein group LFQ intensities were log_2 transformed to reduce the impact of outliers. To reduce quantitative uncertainty, protein groups having less than four valid values (those with MS1 quantification results) present in at least one categorical group, i.e., HRS^WT, HRS^S345A, or HRS^S345D were removed. Missing data points were imputed by creating a downshifted Gaussian distribution of random numbers to simulate the distribution of low signal values (imputation Width = 0.3, shift = 1.8). Perseus was also used for the following analyses: hierarchical clustering (Euclidian distances and k-means clustering) after protein intensity values from all replicates were averaged for each protein within cell type and z score normalized; Principal Component Analysis (PCA) after duplicate LC-MS/MS analyses of the same sample (technical replicates) were averaged; and data visualization using volcano plots. For heat maps, technical replicates were averaged as described above and z score normalized. Ingenuity Pathway Analysis (IPA, Qiagen Inc.; www.qiagenbioinformatics.com/products/ingenuity-pathway-analysis)[64] was used to determine sub-cellular localization. Uniprot protein identifiers for all proteins

identified by LC-MS/MS were uploaded into the application and each identifier was mapped to its corresponding object in Ingenuity's Knowledge Base.

For RPPA data, analysis was performed according to the protocol from the M.D. Anderson Cancer Center. Specifically, relative protein levels for each sample were determined by interpolation of each dilution curves from the "standard curve" (supercurve) of the slide (antibody). Supercurve is constructed by a script in R written by the RPPA core facility. The package binaries of SuperCurve and SuperCurveGUI are available in R-Forge (https://r-forge.r-project.org/R/?group_id=1899). These values are defined as Supercurve Log2 value. All the data points were normalized for protein loading and transformed to linear value, designated as "Normalized Linear". "Normalized Linear" value was transformed to Log2 value, and then median-centered for further analysis. Median-Centered values were centered by subtracting the median of all samples in each protein. All the above-mentioned procedures were performed by the RPPA core facility. The normalized data provided by the RPPA core facility were analyzed by Cluster 3.0 (http://bonsai.ims.u-tokyo.ac.jp/~mdehoon/software/cluster/) and visualized using the Java TreeView 1.0.5 (http://jtreeview.sourceforge.net/).

All other statistical analyses were performed using GraphPad Prism (version 8.0) or Microsoft Excel (office 365, Microsoft). For proteomics analysis, significantly changed proteins for pairwise HRS comparisons were defined as having ≥2-fold change, a permutation-based FDR ≤ 0.05, and $S_0 = 0.1$[65]. Normality of distribution was determined by D'Agostino-Pearson omnibus normality test and variance between groups was assessed by the $F$ test. For IHC staining statistics, non-parametric Mann–Whitney $U$ tests or Wilcoxon matched-pairs tests were used for unpaired and paired analysis, respectively. Correlations were determined by Spearman's $r$ coefficient. One-way ANOVA with Tukey's test was used to the multi-compare differences between vehicle, wild-type HRS, and mutants. Two-way ANOVA with Sidak's test was used for multi-group comparisons. Error bars shown in graphical data represent mean ± s.d. $P < 0.05$ was considered statistically significant.

**Reporting summary**. Further information on research design is available in the Nature Research Reporting Summary linked to this article.

## Data availability

RPPA data are available from the NCBI Gene Expression Omnibus (GEO) under accession number GSE174270. The mass spectrometry proteomics data have been deposited to the MassIVE data repository with the accession number MSV000088846 and to the Proteome Xchange Consortium with the accession number PXD031715. All data are included in the Supplemental Information or available from the authors upon reasonable requests, as are unique reagents used in this article. The raw numbers for charts and graphs are available in the Source Data file whenever possible. Source data are provided in this paper.

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

## Acknowledgements

We are grateful to Dr. Haidong Dong for providing the *PD-L1*-KO B16 cell lines. The work is supported by NIH grants R35 GM141832 to W.G., NCI CA174523 (SPORE) grant to W.G., X.X., R.K.A., M.H., L.M.S., G.C.K., T.C.M., and D.W.S.

## Author contributions

L.G., B.W., and T.L. designed and performed experiments, analyzed data, and wrote the manuscript. L.A.B. performed proteomics and data analysis. G.S., M.L., C.N.L., S.L, C.Y, L.H., and G.S. performed experiments and analyzed data. D.T.F., G.M.B., L.M.S., G.C.K., and T.C.M. collected and analyzed patient samples. K.Q.C. and F.C. performed tissue microarray and immunohistochemistry staining. T.M.S. and M.D. validated data. G.B.M. helped with RPPA study and data analysis. Resources. K.T.F., D.W.S., Y.H.C., and M.H. provided conceptual input to the experiment design and data interpretation. R.K.A. and X.X. provided clinical expertise and input to experiment design, interpretation, and presentation. W.G. supervised the project and wrote the manuscript.

## Competing interests

Wei Guo received research funding from Bristol Meyers Squibb and Exio Inc. for bio-marker studies. The funders played no roles in the conceptualization, design, data collection, analysis, the decision to publish, or preparation of the manuscript.
