## [Peer Review File · Nature Communications]

REVIEWER COMMENTS

Reviewer #1 (Remarks to the Author):

Guan, Wu and Li present an interesting manuscript describing that HRS is phosphorylated at Ser345 by ERK, that this phosphorylation leads to enhanced packaging of the immune-checkpoint molecule PD-L1 into small extracellular vesicles (most likely exosomes). The presence of p-HRS (and/or sEVs from cells in which p-HRS was present) led to impaired infiltration of T-cells into the tumour microenvironment, with concordant blunting of the anti-tumour effect of anti-PD-L1 antibodies on tumour growth. This is an extensive study, the quality of the data is generally high and many elements are new and exciting. I think it will be of interest to the sEV, cancer biology and cell biology fields. I am broadly in favour of publication and but have some suggestions below:

1. The HRS phosphorylation is clear and novel. HRS is more commonly known for being tyrosine-phosphorylated downstream of growth factor receptor activation (its proper name is HGS, Hepatocyte Growth Factor Regulated Tyrosine Kinase Substrate – it would be good to get this acronym in the manuscript somewhere) but I will admit that the mechanistic implications of its tyrosine phosphorylation is unclear. Is the ERK-dependent Ser345 phosphorylation constitutive, or does it occur upon stimulation from mitogens? Does HRS-phosphorylation lead to just PDL1 packaging, or are other ubiquitinated/ESCRT-sorted cargos similarly enhanced in the sEV fraction?
2. The T-cell infiltration difference seems relatively minor (e.g., 1F: 1.9 – 1.4 T cells/mm³), albeit significant. I think 1J could benefit from an R² plot to show correlation between high pS345 and low T-cell infiltrate. There are certainly regions on the IHC where this correlation doesn't seem to hold.
3. I think further discussion of the effects of the S345A mutant is required. Is this having a dominant-negative effect on packaging of PD-L1? A limitation of the study is that the authors consider only overexpression of HRS and mutants without removing the endogenous protein. If the S345A HRS prevents PDL1 packaging into sEVs, does this mean that cell surface PDL1 levels are elevated and if so, does this mean that cell surface PDL1 does not prevent T-cell infiltration?
4. The link between T-cell infiltration and T-cell activation is not clear. Are the authors suggesting that PDL1 on sEVs acts as a chemoattractant, or is exosomal PD-L1 'turning off' T-cells and limiting their ability to respond to chemoattractant signals? I think more mechanistic information around how the sEVs block T-cell infiltration would be valuable.
5. The authors generated PD-L1 k/o cells and used them to explore their regulation of CD8 cytotoxicity, examined T-cell infiltration in tumours formed from PD-L1 k/o cells and the ability of WT and S345D exosomes to promote tumour growth. I might have missed it, but can the authors use these PD-L1 k/o cells to generate sEVs in the WT and S345A/D background and see in the transwell

system whether migration was impaired and in the in-vivo sEV infusion system, whether T-cell infiltration was altered? I think this would be needed to prove a causal link between PD-L1 on the S345D sEVs and T-cell infiltration and might help answer point 4.

6. This is minor, but in 3C, VAPA is an integral ER membrane protein, not a PM protein

Reviewer #2 (Remarks to the Author):

Review of Manuscript#: NCOMMS-21-41954

Corresponding Author: Wei Guo

Title: HRS Phosphorylation Selectively Drives Immunosuppressive Exosome Secretion and Spatially Restricts CD8 Lymphocyte Infiltration into Tumors

Summary

The manuscript is an innovative and comprehensive analysis of the role of HRS phosphorylation, PDL1 incorporation into exosomes, and immune regulation of tumors. The authors very nicely show the correlation between HRS phosphorylation and T cell infiltration in human cancer and murine models, using unique reagents. This is impactful work. The authors continue their analysis to show that HRS phosphorylation generates exosomes that are qualitatively different, and that PDL1 is one of the proteins impacted by HRS. Again, using unique reagents, the authors demonstrate that exosomes from cancer cells can suppress T cell proliferation according to HRS phosphorylation in the cancer cell, and that PDL1 is mechanistic. In vivo studies demonstrate that these exosomes can also impact tumor growth, and the response to PDL1 blockade in vivo. Together, these data are impactful, significant, and comprehensive.

The manuscript is well written and the figures are clear. While there is some confusion that first occurs when following all of the references to the abundance of subfigures and extended data, the presence of these data is important and a strength.

There are some issues that need to be addressed, but alter interpretation not impact.

Major issues.

The manuscript consistently states that pHRS impacts infiltration to tumors, which is consistent with the Figure 1 data linking pHRS with the number of T cells in the tumor. However, the PDL1 mechanism doesn't fit well with these data. Entry of T cells to tumors is not via the margins, but via

diapedesis across vasculature. The interaction between T cells and vasculature is not an antigen-specific interaction influenced by PDL1, but via chemokines and adhesion molecules. If exosomes are attached to extracellular matrix and interacting with T cells as suggested in the manuscript, then these T cells are already inside the tumor. The chemotaxis assays used in the manuscript model this phenomenon – there is not a vascular bed for the cells to cross in the assay, only matrix. Therefore, if PDL1 on exosomes acts on T cells that have already crossed the vasculature and entered the tumor stroma, then this is not a mechanism to prevent T cell infiltration. As mentioned above, this does not greatly alter the impact of the manuscript, but the assumptions about infiltration throughout should be tempered.

Related to the above, the major effect of the exosomes on the T cells is anti-proliferative. This can reasonably result in decreased numbers of these cells in the tumor, unrelated to regulation of infiltration. This can explain many of the data including where treated cells are adoptively transferred.

The fact that HRS phosphorylation acts via PDL1 regulation appears well supported, but there are inconsistencies. Firstly, the major differences on tumor growth are between the HRS S345D mutant and the others. In Figure 2 and Figure 4 the D mutant has similar PDL1 surface expression to the WT, yet these tumors respond differently to PDL1 blockade. Similarly, the HRS S345A mutant has low or absent PDL1 expression, yet behaves very similarly to the WT. These data suggest that PDL1 levels on the exosomes is not the primary mechanism at work for pHRS impact on tumor responses to therapy. These inconsistencies should be addressed.

The source of the mutant cell lines is not given. Specifically, how the cell lines were engineered to exclude wt HRS and express mutant HRS. Is this knockout of host followed by transfection of mutant, or in situ mutation. Plasmids, targeting reagents etc should be described.

For the murine tumor analyses, more information is needed.

1. The timing of tumor harvest should be provided. At present this is only given for the adoptive transfer studies.
2. The start time of each treatment should be provided. The methods are unclear, but suggest PDL1 blockade, exosome treatment, etc starts at the time of tumor implantation into mice, and continue throughout the experiment. This will very much influence whether the treatments can be interpreted as blocking initial anti-tumor T cell responses, or blocking infiltration of T cells to established tumors.
3. Related to the above, the PD1 antibody treatment appears to be continuously given every 3d. This is well inside the half-life of the Ab, will result in higher overall levels throughout the experiment, and can result in anti-Ab responses with long-term administration. If this treatment plan is correct it is not a problem, but it should be clarified so that readers can interpret the data accordingly.
4. The reason that survival studies have not been performed should be explained.
5. The number of times that the murine experiments were repeated should be provided.

Minor issues

The manuscript consistently refers to tumors as xenografts. While the human cancer cells form a xenograft, B16 and Yummi are syngeneic mouse tumors and not xenogenic. This should be corrected throughout.

Representative flow cytometry should be shared in the supplemental figures. The interpretation of this data is important, so examples showing how T cells are gated and phenotyped should be provided. One extra supplement would be sufficient – no need to do this for each experiment.

Very minor format line 471

Reviewer #3 (Remarks to the Author):

In this manuscript, Guan, Wu, Li et al. identify packaging of PD-L1 into exosomes/sEVs mediated by phosphorylation of HRS by ERK. High HRS phosphorylation was associated with CD8 T cell exclusion and resistance to PD-1 blockade. The authors describe a novel mechanism of exosome/sEV-mediated immunosuppression and provide in depth studies to mechanistically support the role of HRS phosphorylation in T cell dysfunction. I have a few minor comments related to the manuscript, as outlined below.

1. The authors demonstrate that HRS mutant cell lines secrete similar numbers of total number of sEVs, as quantified by NTA. The number of sEVs secreted should be normalized by cell number to account for differences in cell proliferation and a secondary quantification of sEV secretion should be included (e.g., sEV protein quantification).

2. sEVs are observed in association with ECM fibers and the authors propose that sEVs are present in the tumor microenvironment to alter T cell responses. While no significant upregulation of ITGA2, ITGA3, and ITGA5 was observed in sEVs from HRS mutant cells, adding a list of integrins detected in sEVs (expressed vs. not expressed, as opposed to relative abundance) would strengthen the argument that sEVs bind to ECM. In addition, EV5j is missing a figure legend.

3. The authors show that sEVs derived from HRS mutant cells can influence T cell activation; however, these experiments are based on treatment with ex vivo isolated sEVs at a single dosage.

Including T cell dose-dependent response of sEV treatment (at least for in vitro experiments) would be helpful to better appreciate the physiological relevance of these findings.

4. Methods section should be updated to include information on the plasmids used for generating cell lines with wild-type and mutant forms of HRS.

We thank all the 3 reviewers for their highly positive comments on the novelty and quality of our work. We are also grateful for their questions and suggestions, which are all constructive. In the revision, we have addressed all of the questions. The following are our point-by-point responses:

Reviewer #1:

“Guan, Wu and Li present an interesting manuscript describing that HRS is phosphorylated at Ser345 by ERK, that this phosphorylation leads to enhanced packaging of the immune-checkpoint molecule PD-L1 into small extracellular vesicles (most likely exosomes). The presence of p-HRS (and/or sEVs from cells in which p-HRS was present) led to impaired infiltration of T-cells into the tumour microenvironment, with concordant blunting of the anti-tumour effect of anti-PD-L1 antibodies on tumour growth. This is an extensive study, the quality of the data is generally high and many elements are new and exciting. I think it will be of interest to the sEV, cancer biology and cell biology fields. I am broadly in favour of publication and but have some suggestions below:

1. The HRS phosphorylation is clear and novel. HRS is more commonly known for being tyrosine-phosphorylated downstream of growth factor receptor activation (its proper name is HGS, Hepatocyte Growth Factor Regulated Tyrosine Kinase Substrate – it would be good to get this acronym in the manuscript somewhere) but I will admit that the mechanistic implications of its tyrosine phosphorylation is unclear. Is the ERK-dependent Ser345 phosphorylation constitutive, or does it occur upon stimulation from mitogens? Does HRS-phosphorylation lead to just PDL1 packaging, or are other ubiquitinated/ESCRT-sorted cargos similarly enhanced in the sEV fraction?”

Thanks for raising the question. As the reviewer pointed out, HRS is also known as HGS (*Hepatocyte Growth Factor Regulated Tyrosine Kinase Substrate*). In the membrane trafficking field, the term “HRS” is more commonly used. HGS/HRS was originally discovered as hepatocyte growth factor regulated tyrosine kinase substrate, but its phosphorylation on tyrosine is much less investigated. Following reviewer’s suggestion, we included the name HGS in the revised Introduction.

The phosphorylation of Ser345 reported in the current study is different from the tyrosine phosphorylation of HRS. It is well established that the ERK is constitutively activated in metastatic melanoma due to prevailing mutations such as *BRAF^{V600E}*. We observed constitutive phosphorylation on HRS on Ser345 in melanoma tissues (Fig. 3; Extended Data Fig. 1 and 2).

HRS phosphorylation enhanced the packaging of a specific group of cargo including, but not limited to, PD-L1 (Fig.3c & d; Extended Data Fig. 5g). On the other hand, conventional cargo proteins such as CD9, CD81, GPCRs, TGF- β receptors, IGFRs and integrins, were not affected (Fig. 3d; Extended Data Fig. 5j). HRS is thought to bind to ubiquitinated cargos such as E-Cadherin through its ubiquitin-interacting motif (UIM). Our domain mapping analysis showed that PD-L1 interacts with HRS through a.a. 275-478, which is different from the UIM (Fig. 3i and 3j). Given that S345 is located within the region for PD-L1 binding, the data suggest that S345 phosphorylation specifically modulates ubiquitin-independent sorting.

“2. The T-cell infiltration difference seems relatively minor (e.g., 1F: 1.9 – 1.4 T cells/mm³), albeit significant. I think 1J could benefit from an R² plot to show correlation between high pS345 and low T-cell infiltrate. There are certainly regions on the IHC where this correlation doesn’t seem to hold.”

Thanks for the suggestion! Due to the heterogeneity and complexity of human samples, the correlation may not apply to all areas of the tumors (e.g. areas that are about to undergo apoptosis or necrosis). Furthermore, HRS is not phosphorylated in every cell of the tumor clusters. For Fig.1J, we indeed used R^2 . In Extended Data Fig. 2h, we compared the number of T cells in the regions with high or low levels of pHRS within the tumors. Regions with high pHRS showed a significantly smaller number of T cells.

“3. I think further discussion of the effects of the S345A mutant is required. Is this having a dominant-negative effect on packaging of PD-L1? A limitation of the study is that the authors consider only overexpression of HRS and mutants without removing the endogenous protein. If the S345A HRS prevents PDL1 packaging into sEVs, does this mean that cell surface PDL1 levels are elevated and if so, does this mean that cell surface PDL1 does not prevent T-cell infiltration?”

Thanks for raising the question. We agree with the reviewer that the observed effect was dominant-negative for cells expressing the phospho-deficient S345A-HRS mutant. In the revised manuscript, we have indicated the dominant-negative nature of the mutant in Discussion. In addition, we provided more of our reasoning in the Results section when describing specific experiments involving the mutant (in red).

The blocking of PD-L1 packaging into sEVs in the S345A-HRS mutant did not significantly affect the levels of PD-L1 on cell surface (Extended Data Fig.5 a-d). As shown in our IHC staining (as well as others shown in literature), most tumor cells did not directly interact with T cells. Therefore, PD-L1 on surface of these tumor cells may not contribute directly to the T cell inhibition. This further underscores the importance of the tumor microenvironment, particularly sEV PD-L1 in this study, on T cell infiltration.

“4. The link between T-cell infiltration and T-cell activation is not clear. Are the authors suggesting that PDL1 on sEVs acts as a chemoattractant, or is exosomal PD-L1 ‘turning off’ T-cells and limiting their ability to respond to chemoattractant signals? I think more mechanistic information around how the sEVs block T-cell infiltration would be valuable.”

Based on the data, we propose that PD-L1 on sEVs acts to “turn off” T cells. As shown in Fig. 4d-f, sEVs derived from cells expressing HRS^{WT} or HRS^{S345D} inhibited T cell proliferation and activation (as assayed by Ki-67 and GzmB expression, and proteomics profiling). Mechanistically, the inhibitory effect was carried out through PD-L1-mediated checkpoint function, as PD-L1 knockout or antibody blocking attenuated the suppression effect (Figure 5). Please also see below on Point 5.

“5. The authors generated PD-L1 k/o cells and used them to explore their regulation of CD8 cytotoxicity, examined T-cell infiltration in tumours formed from PD-L1 k/o cells and the ability of WT and S345D exosomes to promote tumour growth. I might have missed it, but can the authors use these PD-L1 k/o cells to generate sEVs in the WT and S345A/D background and see in the transwell system whether migration was impaired and in the in-vivo sEV infusion system, whether T-cell infiltration was altered? I think this would be needed to prove a causal link between PD-L1 on the S345D sEVs and T-cell infiltration and might help answer point 4.”

We have performed experiments and found that PD-L1 KO in cells of different HRS mutant backgrounds abolished the T cell inhibitory effects of their sEVs in both the transwell assay *in vitro* and the adoptive T cell infiltration *in vivo*. The data is now included in Extended Data Figure 13a and b. Thanks for the suggestion!

“6. This is minor, but in 3C, VAPA is an integral ER membrane protein, not a PM protein”.

Thanks for pointing this out. VAPA was listed as a PM protein in the database of Uniprot. To avoid confusion, we have now removed it from the list.

Again, we thank Reviewer #1 for the strong support and constructive suggestions!

Reviewer #2:

“Review of Manuscript#: NCOMMS-21-41954

Corresponding Author: Wei Guo

Title: HRS Phosphorylation Selectively Drives Immunosuppressive Exosome Secretion and Spatially Restricts CD8 Lymphocyte Infiltration into Tumors

Summary

The manuscript is an innovative and comprehensive analysis of the role of HRS phosphorylation, PDL1 incorporation into exosomes, and immune regulation of tumors. The authors very nicely show the correlation between HRS phosphorylation and T cell infiltration in human cancer and murine models, using unique reagents. This is impactful work. The authors continue their analysis to show that HRS phosphorylation generates exosomes that are qualitatively different, and that PDL1 is one of the proteins impacted by HRS. Again, using unique reagents, the authors demonstrate that exosomes from cancer cells can suppress T cell proliferation according to HRS phosphorylation in the cancer cell, and that PDL1 is mechanistic. In vivo studies demonstrate that these exosomes can also impact tumor growth, and the response to PDL1 blockade in vivo. Together, these data are impactful, significant, and comprehensive.

The manuscript is well written and the figures are clear. While there is some confusion that first occurs when following all of the references to the abundance of subfigures and extended data, the presence of these data is important and a strength.

There are some issues that need to be addressed, but alter interpretation not impact.

Major issues.

The manuscript consistently states that pHRS impacts infiltration to tumors, which is consistent with the Figure 1 data linking pHRS with the number of T cells in the tumor. However, the PDL1 mechanism doesn't fit well with these data. Entry of T cells to tumors is not via the margins, but via diapedesis across vasculature. The interaction between T cells and vasculature is not an antigen-specific interaction influenced by PDL1, but via chemokines and adhesion molecules. If exosomes are attached to extracellular matrix and interacting with T cells as suggested in the manuscript, then these T cells are already inside the tumor. The chemotaxis assays used in the manuscript model this phenomenon – there is not a vascular bed for the cells to cross in the assay, only matrix. Therefore, if PDL1 on exosomes acts on T cells that have already crossed the vasculature and entered the tumor stroma, then this is not a mechanism to prevent T cell infiltration. As mentioned above, this does not greatly alter the impact of the manuscript, but the assumptions about infiltration throughout should be tempered. Related to the above, the major effect of the exosomes on the T cells is anti-proliferative. This can reasonably result in decreased numbers of these cells in the tumor, unrelated to regulation of infiltration. This can explain many of the data including where treated cells are adoptively transferred.”

Thanks for raising this point. Increased lymphocyte density at the tumor periphery has been observed in many studies, suggesting that lymphocytes may enter tumor through the invasive front. High stromal CD8⁺ T-cell density at the tumor periphery and high parenchymal CD8⁺ T-cell density at the invading edge were independent prognostic makers. Thus, tumor infiltrating lymphocytes are likely coming from two different sources, infiltrating through the tumor periphery and diapedesis. Recent studies indicate that patients with high intratumoral, but not peritumoral, CD8⁺ T cells, had a better response to immune checkpoint blockade-based therapies. Vasculature is present in the tumor stroma. Lymphocytes migrate out of vasculature and enter tumor stroma and these lymphocytes do not automatically become tumor infiltrating lymphocytes (TILs) that are in direct contact with tumor cells. We have observed melanoma cases with perivascular lymphocytes in the stroma but rare TILs, and showed previously that stromal inflammatory cells were associated with poorer prognosis in the melanoma (PMID: 30965022). For stromal lymphocytes to become TILs, they have to overcome the barrier between tumor stromal and tumor parenchyma. Our data supports the hypothesis that exosomes secreted by melanoma cells contribute to this barrier and confine lymphocytes in the tumor stroma.

“The fact that HRS phosphorylation acts via PDL1 regulation appears well supported, but there are inconsistencies. Firstly, the major differences on tumor growth are between the HRS S345D mutant and the others. In Figure 2 and Figure 4 the D mutant has similar PDL1 surface expression to the WT, yet these tumors respond differently to PDL1 blockade. Similarly, the HRS S345A mutant has low or absent PDL1 expression, yet behaves very similarly to the WT. These data suggest that PDL1 levels on the exosomes is not the primary mechanism at work for pHRS impact on tumor responses to therapy. These inconsistencies should be addressed.”

Thanks for raising this question and giving us the opportunity to clarify our points. We observed no difference in the expression of PD-L1 on the surface of cells expressing HRS wild type, S345A or S345D (Extended Data Fig. 5a and b). However, PD-L1 levels were significantly higher on sEV^{S345D} compared to the sEV^{WT} and sEV^{S345A} (Extended Data Fig. 6a), consistent with the data that HRS phosphorylation up-regulated the loading of PD-L1 onto the exosomes (Fig. 3a-3j; Extended Data Fig. 6). While the same level of cell surface PD-L1 expression suggests that the tumor cells have the equal *potential* to respond to anti-PD-1 treatment, the difference is that their sEVs carry different amounts of PD-L1, and both our *in vitro* and *in vivo* assays (Figure 4 and 5 and related supplementary figures) demonstrate that these sEVs differed in their ability in T cell inhibition. In other words, the tumor microenvironment is different as sEVs from S345D mutant potentially prevented T cell infiltration, whereas those from S345A mutant did not show such effect. Thus, despite the same PD-L1 cell surface expression, the spatial block of T cell infiltration by exosomes leads to different response to anti-PD-1 antibody treatment.

“The source of the mutant cell lines is not given. Specifically, how the cell lines were engineered to exclude wt HRS and express mutant HRS. Is this knockout of host followed by transfection of mutant, or in situ mutation. Plasmids, targeting reagents etc should be described.”

We have now included more information in Methods section of the revised manuscript.

“For the murine tumor analyses, more information is needed.

1. The timing of tumor harvest should be provided. At present this is only given for the adoptive transfer studies.”

We have included more information on the timing of tumor harvest in Methods (page 18).

“2. The start time of each treatment should be provided. The methods are unclear, but suggest

PDL1 blockade, exosome treatment, etc starts at the time of tumor implantation into mice, and continue throughout the experiment. This will very much influence whether the treatments can be interpreted as blocking initial anti-tumor T cell responses, or blocking infiltration of T cells to established tumors.”

Thanks for the suggestion. We have now provided more detailed information it in the Method session (Page 18).

“3. Related to the above, the PD1 antibody treatment appears to be continuously given every 3d. This is well inside the half-life of the Ab, will result in higher overall levels throughout the experiment, and can result in anti-Ab responses with long-term administration. If this treatment plan is correct it is not a problem, but it should be clarified so that readers can interpret the data accordingly.”

Thanks for raising this point. The PD-1 antibody treatment followed the regime described previously (PMID: 33458695, PMID: 28379630, PMID: 26359984). We have now provided more detailed information it in the Method session and referenced PMID: 26359984 in Page 18.

“4. The reason that survival studies have not been performed should be explained.”

We needed to collect the tumors of different groups at the same time for flow cytometry and other experimental analysis. Therefore, we were not able to conduct survival study in this case.

“5. The number of times that the murine experiments were repeated should be provided.”

We have now provided the information in Figure legends (Page 31, 33, 34).

“Minor issues

The manuscript consistently refers to tumors as xenografts. While the human cancer cells form a xenograft, B16 and Yummi are syngeneic mouse tumors and not xenogenic. This should be corrected throughout.”

Thanks for pointing this out. We have now replaced the term with Tumor tissue derived EVs (“TTDE”) throughout the revised manuscript.

“Representative flow cytometry should be shared in the supplemental figures. The interpretation of this data is important, so examples showing how T cells are gated and phenotyped should be provided. One extra supplement would be sufficient – no need to do this for each experiment.”

Thank you for the suggestion. We have now included representative flow cytometry data in Extended Data Fig 8.

“Very minor format line 471”

Thanks for pointing out the error. We have made the correction in the revised version.

We thank Reviewer #2 for the positive comments and constructive suggestions that helped us improve the paper!

Reviewer #3:

“In this manuscript, Guan, Wu, Li et al. identify packaging of PD-L1 into exosomes/sEVs

mediated by phosphorylation of HRS by ERK. High HRS phosphorylation was associated with CD8 T cell exclusion and resistance to PD-1 blockade. The authors describe a novel mechanism of exosome/sEV-mediated immunosuppression and provide in depth studies to mechanistically support the role of HRS phosphorylation in T cell dysfunction. I have a few minor comments related to the manuscript, as outlined below.

1. The authors demonstrate that HRS mutant cell lines secrete similar numbers of total number of sEVs, as quantified by NTA. The number of sEVs secreted should be normalized by cell number to account for differences in cell proliferation and a secondary quantification of sEV secretion should be included (e.g., sEV protein quantification)."

Thanks for raising this question. Cells expressing different HRS variants showed no difference in their rate of proliferation, as examined using CCK-8 assay:

In our sEV collection experiments, we plated equal numbers of cells at the beginning and the final number was confirmed equal. Also, consistent with the reviewer's suggestion, for the comparison of the levels of PD-L1 on sEVs, we used the same amounts of sEVs on western blotting based on total sEV protein quantification as indicated in the original manuscript (Page 31).

"2. sEVs are observed in association with ECM fibers and the authors propose that sEVs are present in the tumor microenvironment to alter T cell responses. While no significant upregulation of ITGA2, ITGA3, and ITGA5 was observed in sEVs from HRS mutant cells, adding a list of integrins detected in sEVs (expressed vs. not expressed, as opposed to relative abundance) would strengthen the argument that sEVs bind to ECM. In addition, ED5j is missing a figure legend."

The following is our analysis of different ITGAs:

-We did not list the expression of integrins in the manuscript as they all bind to ECM. Showing the expression of different integrins could distract the readers from the main thesis of the paper.
-We added legend for Extended Data Fig 5j in the revision. Thank you so much for pointing out our error!

“3. The authors show that sEVs derived from HRS mutant cells can influence T cell activation; however, these experiments are based on treatment with ex vivo isolated sEVs at a single dosage. Including T cell dose-dependent response of sEV treatment (at least for in vitro experiments) would be helpful to better appreciate the physiological relevance of these findings.”

Thanks for the suggestion. We have now included the dose-dependent response of sEV treatment for the *in vitro* experiment in Extended Data Fig. 8c and d.

“4. Methods section should be updated to include information on the plasmids used for generating cell lines with wild-type and mutant forms of HRS.”

We have now included the information in Method section of the revised manuscript.

We thank Reviewer #3 for the strong support and constructive suggestions that help us improve the paper!

REVIEWERS' COMMENTS

Reviewer #1 (Remarks to the Author):

The authors have made a good attempt to address concerns I raised in review and I am happy with their responses. I think the data are novel and exiting.

One minor point re data presentation - is it possible to get the colourful key for Figure 4d and 4e more appropriately placed? Currently half of it is under 4d and half of it is under 4e and it looks like they are bar-labels, but they aren't.

Reviewer #2 (Remarks to the Author):

The authors have satisfactorily answered this reviewer's concerns. While I disagree on the mechanism of T cell entry to tumors - recirculation and vascular entry to tumors remains the only proven mechanism of entry, and transmigration from the invasive front to the tumor core has not been mechanistically proven - the authors have given fair responses.

There are no further major or minor concerns from this reviewer.

Reviewer #3 (Remarks to the Author):

The authors have revised the manuscript to show that cancer cell proliferation is not impacted by HRS mutant expression, provide dose-dependent response to sEV treatment, and include additional details in the methods section. All of my comments have been adequately addressed.

Response to reviewers:

Reviewer #1:

The authors have made a good attempt to address concerns I raised in review and I am happy with their responses. I think the data are novel and exiting.

One minor point re data presentation - is it possible to get the colourful key for Figure 4d and 4e more appropriately placed? Currently half of it is under 4d and half of it is under 4e and it looks like they are bar-labels, but they aren't.

Response: Thanks for raising this point. We have moved the color keys to the right side of the figures, as with the labeling of other figures.

Reviewer #2:

The authors have satisfactorily answered this reviewer's concerns. While I disagree on the mechanism of T cell entry to tumors - recirculation and vascular entry to tumors remains the only proven mechanism of entry, and transmigration from the invasive front to the tumor core has not been mechanistically proven - the authors have given fair responses.

There are no further major or minor concerns from this reviewer.

Response: We appreciate reviewer's point, and are grateful for the constructive suggestions.

Reviewer #3:

The authors have revised the manuscript to show that cancer cell proliferation is not impacted by HRS mutant expression, provide dose-dependent response to sEV treatment, and include additional details in the methods section. All of my comments have been adequately addressed.

Response: We are grateful for the reviewer's constructive suggestions that helped us improve the paper.